# LLM Reasoning for Machine Translation: Synthetic Data Generation over Thinking Tokens

## Abstract

Large reasoning models (LRMs) have led to new possibilities in terms of problem-solving, through the devising of a natural language thought process prior to answering a query. While their capabilities are well known across mathematics and coding tasks, their impact on the task of machine translation (MT) remains under-explored. In this work, we explore the benefits of the generation of intermediate tokens when performing MT across multiple language pairs of different levels of resourcedness and multiple setups. We find that "thinking tokens" do not help LRMs better perform MT. This result generalizes to models fine-tuned to reason before translating using distilled chain of thought (CoT) inspired by human translators' practices. Specifically, fine-tuning a model with synthetic CoT explanations detailing how to translate step-by-step does not outperform standard input-output fine-tuning. However, constructing the CoT based on MT prompting strategies results in improvements. Our findings underscore that the contribution of a CoT during fine-tuning highly depends on the presence of translation attempts in them. More broadly, our results suggest that using a teacher to refine target translations or to expand parallel corpora is more impactful than distilling their CoT explanations into "thinking" MT models.

## 1 Introduction

Large Language Models (LLMs) are general-purpose problem solvers (Touvron et al., 2023; OpenAI et al., 2024; Dubey et al., 2024; Kimi Team et al., 2025). Their instruction-following capabilities help them carry out a wide variety of requests provided by users via text. Research on alignment, typically through Reinforcement Learning from Human Feedback (RLHF) (Askell et al., 2021; Bai et al., 2022; Ouyang et al., 2022; Rafailov et al., 2023; Lambert et al., 2025) has greatly contributed to improving the quality of LLMs' outputs. Recently, a new paradigm has emerged: to train LLMs to "think" in natural language before answering a query. OpenAI o1 and o3 (OpenAI, 2024), DeepSeek-R1 (DeepSeek-AI et al., 2025), Qwen3 (Yang et al., 2025), Claude 4 (Anthropic, 2025) and Gemini 2.5 (Gemini Team et al., 2025) *inter alia* are instances of these Reasoning Models (RM) or Thinking Models (TM). They capitalize on RL to generalize the success of Chain-of-Thought (CoT) prompting (Wei et al., 2022) during training for improved safety robustness and performance. They particularly excel in reasoning-intensive tasks such as olympiad-level mathematics (AIME 2024/2025, HMMT etc.) and competition-level coding (Shi et al., 2024; Quan et al., 2025). When it comes to Machine Translation (MT), they also perform well (Chen et al., 2025) notably for stylized translation and document-level MT (Liu et al., 2025).

Labeling o1-like models as *thinking* presupposes that the intermediate tokens they produce before answering meaningfully reflect their reasoning process. Many works challenge this view. For instance, Bhambri et al. (2025) find little correlation between the correctness of final answers and the accuracy of intermediate traces, echoing earlier results by Turpin et al. (2023) showing that CoT explanations can be unfaithful. Ma et al. (2025) further demonstrate that "not thinking" can outperform "explicit reasoning" on certain challenging tasks, motivating exploration of "not thinking" in other settings. In this work, we examine the value of intermediate tokens in MT with LLMs and ask what forms of intermediate information are actually beneficial. MT is a particularly interesting

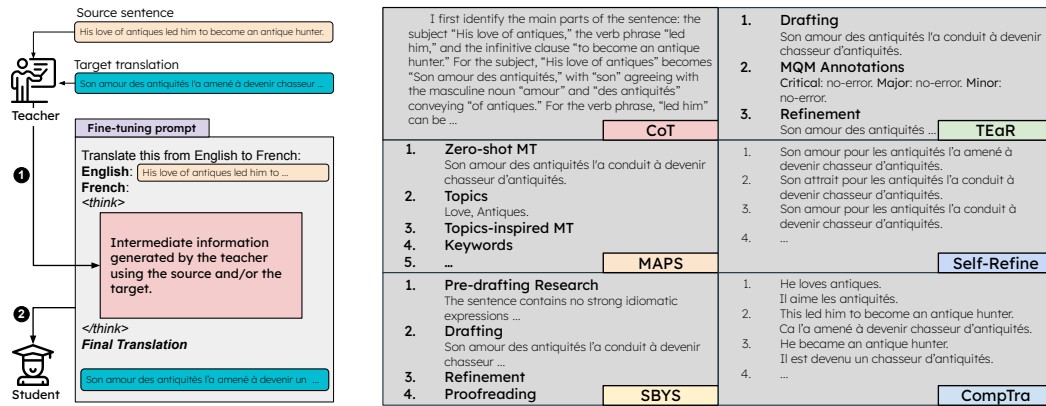

Figure 1: CoT FINE-TUNING (left): Given a source-target pair, a teacher is prompted to get a thought process on how to obtain the target given the source based on a given strategy (right). The obtained trace is used as intermediate information to fine-tune a student to "think" before translating.

task in this context as CoT prompting has been shown not to result in better translation than vanilla few-shot prompting (Nguyen & Xu, 2025). We show that:

- **The thinking mode of LRMs does not results in better MT outputs.** We carry out extensive experiments across ten language directions and find no significant benefit from prior thinking. Our experiments cover zero-shot and few-shot settings, three benchmarks, high- and low-resource language pairs (X-to-English and English-to-X directions) and different temperatures of generation, and they all point to the same conclusion.

- **CoT distillation does not outperform standard fine-tuning.** Many works report a stark improvement of reasoning abilities when fine-tuning a small model to think before answering, using the CoT outputs of a teacher (Zelikman et al., 2022; Huang et al., 2024; Li et al., 2025; Guha et al., 2025). We apply this setup to fine-tune LMs to "think" before translating and compare it to standard input-output fine-tuning. Our experiments with `gemma-3-4b-pt` on English to Xhosa suggest the consistent superiority of standard fine-tuning across six different MT-specific CoT templates.

- **Using traces obtained by translating the source with modular prompting strategies specifically designed for MT outperforms CoT distillation and standard input-output fine-tuning, but ultimately data matters most.** Instead of vanilla CoT distillation, we propose to use the traces obtained when the teacher attempts to translate the source using a modular prompting strategy for MT. Such strategies typically involve an analysis of the source, the proposal of intermediate candidates, and the derivation of the final translation. These steps can be concatenated into a single text, which we then use as intermediate information for fine-tuning the student model. This approach outperforms input-output fine-tuning by up to 3.5 BLEU and 2 MetricX points. Analysis indicates that the gains stem mainly from translation attempts embedded in the traces. We further show that using the teacher to improve the fine-tuning dataset instead, by either enhancing the quality of its target translations or generating additional parallel pairs has greater benefits than relying on thinking tokens, without incurring extra inference cost after fine-tuning.

## 2 RELATED WORK

**Reasoning with LLMs.** CoT prompting (Wei et al., 2022) has revolutionized the approach to reasoning with LLMs. Following In-Context Learning (ICL;[1] Brown et al., 2020), CoT prompting drives the LLM to explain with natural language the thought process before deriving the solution to a problem. It was shown to be particularly useful for mathematical tasks that require the LLM to

---

[1]Also referred to as *few-shot learning*, which is the ability through which LLMs can carry out a wide variety of tasks at inference based on a few demonstrations

think through a set of reasoning steps (Cobbe et al., 2021; Hendrycks et al., 2021a;b; Suzgun et al., 2023). The intuition behind CoT prompting and its success has powered countless related prompting strategies (Kojima et al., 2022; Zhang et al., 2023; Yasunaga et al., 2024). Other developments involve using CoT as a building block to solve sequential problems (Zhou et al., 2023; Zebaze et al., 2024), using CoT in combination with an external tool such as a programming language interpreter (Chen et al., 2023; Gao et al., 2023) or to reason on diverse reasoning trajectories (Wang et al., 2023; Yao et al., 2023; Besta et al., 2024; Bi et al., 2025). CoT-based techniques have also been used to create datasets for supervised fine-tuning (Zelikman et al., 2022; Shao et al., 2024; Yue et al., 2024), which is often subsequent to prior continual pretraining on mathematics and code data (Lewkowycz et al., 2022; Azerbayev et al., 2024). We now have Thinking Models which are trained to "think" and produce a long chain of thought before responding to a user query, and these models have set a new state-of-the-art for multiple benchmarks (GPQA (Rein et al., 2024); SWE-Bench (Jimenez et al., 2024; Chowdhury et al., 2024)) and keep motivating the creation of more challenging ones (Phan et al., 2025; Chollet et al., 2025). Despite their success, Ma et al. (2025) suggest that these thinking tokens can bring limited gains compared to not thinking[2] on some reasoning tasks under thinking budget constraints. Others question the informativeness of CoT traces, showing that even incorrect traces can yield correct outcomes (Turpin et al., 2023; Bhambri et al., 2025) and that fine-tuning on incorrect traces can be as effective as on correct ones (Stechly et al., 2025). We focus this debate on the specific task of MT and ask whether general-purpose RMs translate better with thinking tokens or if they can be removed entirely.

**Machine Translation with LLMs.** MT is one of the many tasks that LLMs can perform via ICL. Historically, encoder-decoder models have been the go-to architecture (Sutskever et al., 2014; Cho et al., 2014; Bahdanau et al., 2016; Johnson et al., 2017; Vaswani et al., 2017). Decoder-based LLMs perform on par with or better than supervised MT models such as NLLB (Costa-jussà et al., 2022) when dealing with the so-called high-resource languages (HRLs), largely thanks to the availability of large quantities of high quality data on the internet, which facilitates their incorporation in the ever growing pretraining corpora of LLMs. LLMs still struggle with translating from and into low-resource languages (LRLs), but they offer more flexibility when prompting. ICL and the use of few-shot examples (including their selection and order, their number and quality) greatly impact the quality of MT outputs (Agrawal et al., 2023; Moslem et al., 2023; Hendy et al., 2023; Bawden & Yvon, 2023; Mu et al., 2023; Zhu et al., 2024; Bouthors et al., 2024; Zebaze et al., 2025c), and various prompting strategies have also been developed for MT such as "Multi-Aspect Prompting and Selection" (MAPS; He et al., 2024), "Translating Step-by-Step" (SBYS; Briakou et al., 2024), "Translate, Estimate, and Refine" (TEaR; Feng et al., 2025b), and "Compositional Translation" (CompTra; Zebaze et al., 2025a). This includes strategies that iteratively guide LLMs to refine translations, with or without external feedback (Chen et al., 2024; Xu et al., 2024b; Ki & Carpuat, 2024), inspired by the success of similar approaches in reasoning tasks (Madaan et al., 2023; Shinn et al., 2024). However, standard CoT prompting (e.g., *Let's think step by step*) has had little to no success in LLM-based MT, with most works reporting worse results than standard input-output prompting (Peng et al., 2023; Zebaze et al., 2025a; Nguyen & Xu, 2025). Several works have explored building RMs for MT (Wang et al., 2025a;b;c; He et al., 2025) by closely following what is done for reasoning tasks. They typically prompt a large model (e.g., DeepSeek-R1) with a curated CoT prompt that guides it from the source sentence to the target translation, then use the generated CoT for supervised fine-tuning (SFT) followed by RL fine-tuning. However, Zheng et al. (2025) suggests that thinking does do not help MT performance when applying GRPO (Shao et al., 2024) on rewards that only evaluate the final translation. In this work, we focus on SFT and study the pertinence of thinking tokens via CoT distillation, which is already successful on reasoning tasks (Huang et al., 2024; Li et al., 2025; Guha et al., 2025). Moreover, given a source, we propose to use modular prompting strategies for MT that have been shown to outperform zero- and/or few-shot MT and ask a teacher to translate the source with the strategies. These prompting strategies resemble a reasoning pipeline, decomposed into multiple steps, each guided by a distinct prompt serving a specific purpose (e.g. identification of idiomatic expressions, generation of a similar sentence, quality estimation, or drafting). For instance, MAPS (He et al., 2024) prompts the LLM to analyze the source and extract three translation-related aspects—keywords, topics, and relevant demonstrations—each used to generate a candidate translation, with the final output selected from these and

---

[2]They induce the "not thinking mode" by stopping the thinking process before it even occurs, for example by writing a sentence such as "Okay, I think I have finished thinking".

the zero-shot attempt. SBYS (Briakou et al., 2024) begins with a pre-drafting research step, where the LLM identifies idiomatic or otherwise challenging expressions in the source. Based on this analysis, it is then prompted to produce an initial draft translation, followed by a refinement stage. In a subsequent conversation, the LLM is instructed to proofread the refined translation—reflecting on both the source and the previously generated draft. In TEaR (Feng et al., 2025b), the LLM first produces a draft translation in a few-shot setting, then generates MQM-style error annotations and refines the draft accordingly. Self-Refine (Chen et al., 2024) involves drafting an initial translation and iteratively improving it through self-feedback. In CompTra (Zebaze et al., 2025a), the LLM decomposes the source into smaller phrases, translates them independently in a few-shot manner, and uses these synthetic pairs as additional demonstrations to improve the final translation. The modularity of these methods lies in their multi-step structure. The outputs (or traces) of these individual steps can be concatenated into a single text and used as a CoT (intermediate tokens) to fine-tune a student, thereby building a thinking MT model, i.e. a model that thinks before translating.

## 3 BENCHMARKING LRMS AT SCALE: TO THINK, OR NOT TO THINK?

We first investigate the influence of prior reasoning on the translation quality in general-purpose thinking models. We compare two conditions: (i) the model is allowed to generate reasoning tokens prior to producing the translation (up to 3500 tokens), and (ii) reasoning is explicitly suppressed by appending `<think>\n\n</think>` to the prompt (Non-Thinking Mode; Yang et al., 2025). We carry out experiments with the Qwen3 model family (Yang et al., 2025), ranging in size from 0.6B to 32B parameters, in a zero-shot English-to-X setting for ten FLORES-200 languages: Czech, Finnish, French, German, Japanese, Kazakh, Lithuanian, Portuguese, Spanish, and Turkish. The results, summarized in Table 1 show that the performance with and without prior thinking is similar. Non-thinking is slightly better, in particular in terms of MetricX but the difference is usually less than 0.5 MetricX point. We provide additional results with more models and directions in Appendix B, and with two other benchmarks, NTREX 128 (Federmann et al., 2022; Barrault et al., 2019) and TICO-19 (Anastasopoulos et al., 2020), in Appendix B.3.

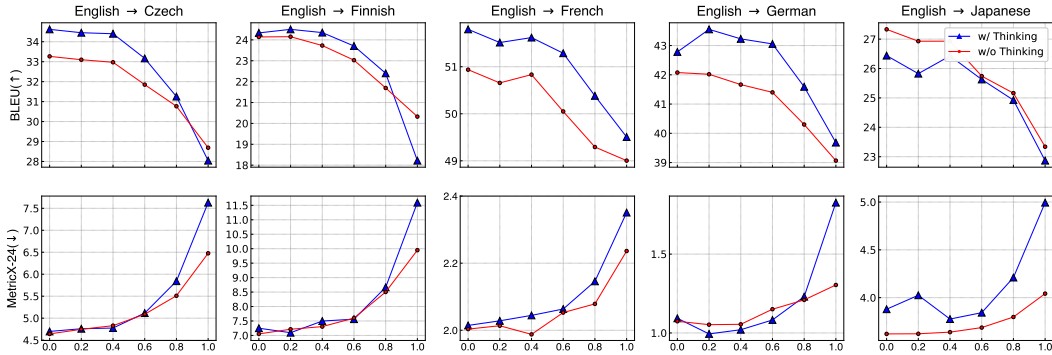

Figure 2: Impact of the Temperature on the translation quality with and without thinking tokens.

We evaluate `Qwen3-32B` when sampling with temperatures 0.2, 0.4, 0.6, 0.8 and 1.0 and plot the results in Figure 2. As opposed to default usage recommendations for TMs, sampling degrades the performance, confirming the results obtained by Chen et al. (2025) with `DeepSeek-R1-670B`. In addition, at each temperature, outputting thinking tokens or not gives approximately the same level of performance. We observe a slight performance gap in favor of thinking tokens with respect to BLEU ($\leq$ 1 BLEU point), but they are behind with respect to MetricX. Low temperatures correlate with higher performance and in particular $T = 0.0$ works best in general. The results of these experiments suggest that MT does not significantly benefit from the presence of thinking tokens. This conclusion is further supported by our human evaluation results in Appendix B.2.

| Models | Czech | | Finnish | | French | | German | | Japanese | |
|---|---|---|---|---|---|---|---|---|---|---|
| | BLEU | MetricX | BLEU | MetricX | BLEU | MetricX | BLEU | MetricX | BLEU | MetricX |
| QWEN3-0.6B | **4.63** | 22.86 | 2.23 | 23.64 | **24.65** | 9.03 | **16.00** | 9.95 | **7.65** | 8.67 |
| *w/o Thinking* | 3.99 | **13.60** | **2.80** | **21.16** | 23.33 | **8.52** | 15.27 | **8.21** | 5.71 | **7.56** |
| QWEN3-1.7B | 14.73 | 15.58 | 5.83 | 20.29 | 37.98 | 4.80 | 27.86 | 4.50 | **15.49** | 5.84 |
| *w/o Thinking* | **15.51** | **15.09** | **7.08** | **19.38** | **38.08** | **4.44** | **28.03** | **3.97** | 15.08 | **5.87** |
| QWEN3-4B | 23.82 | **9.05** | 13.47 | **14.22** | **45.40** | 3.19 | **35.06** | 2.58 | 19.87 | 5.06 |
| *w/o Thinking* | **24.43** | 9.27 | **13.59** | 14.30 | 44.69 | **3.14** | 34.64 | **2.48** | **20.42** | **4.82** |
| QWEN3-8B | 30.11 | **6.38** | **19.29** | 9.95 | 48.72 | 2.59 | **39.29** | 1.62 | 23.45 | 4.31 |
| *w/o Thinking* | **30.27** | 6.61 | 19.21 | 10.20 | **49.03** | **2.49** | 39.05 | **1.56** | **24.43** | **4.08** |
| QWEN3-14B | **34.07** | 4.99 | 22.73 | 7.74 | 51.21 | 2.26 | **42.39** | 1.29 | 26.25 | 3.80 |
| *w/o Thinking* | 33.55 | 5.16 | **23.17** | **7.67** | **51.88** | **2.12** | 41.64 | **1.16** | **27.31** | **3.75** |
| QWEN3-32B | **34.62** | 4.70 | **24.33** | 7.25 | **51.80** | 2.01 | **42.78** | 1.09 | 26.44 | 3.88 |
| *w/o Thinking* | 33.27 | **4.64** | 24.14 | **7.05** | 50.94 | **2.00** | 42.08 | **1.07** | **27.33** | **3.62** |
| Models | Kazakh | | Lithuanian | | Portuguese | | Spanish | | Turkish | |
| | BLEU | MetricX | BLEU | MetricX | BLEU | MetricX | BLEU | MetricX | BLEU | MetricX |
| QWEN3-0.6B | 0.41 | 23.61 | 1.28 | 24.50 | **25.58** | 9.11 | **17.80** | 7.67 | 5.88 | 21.66 |
| *w/o Thinking* | **0.49** | **22.26** | **1.77** | **23.78** | 23.31 | **8.68** | 17.00 | **7.04** | **6.27** | **20.00** |
| QWEN3-1.7B | 0.92 | **23.41** | 5.03 | **21.70** | **39.02** | 4.46 | 25.43 | 3.87 | 13.20 | 15.48 |
| *w/o Thinking* | **1.36** | 23.54 | **5.83** | 21.56 | 39.01 | **4.35** | **25.45** | **3.75** | **14.35** | **13.94** |
| QWEN3-4B | 8.02 | **16.02** | 12.83 | 15.69 | **46.08** | 2.97 | 28.46 | **2.69** | 21.49 | 9.75 |
| *w/o Thinking* | **8.15** | 16.37 | **13.05** | **15.45** | 45.80 | 3.09 | **28.70** | **2.69** | **21.74** | **9.55** |
| QWEN3-8B | **13.76** | **11.68** | 17.49 | 11.51 | **48.93** | 2.53 | **30.77** | 2.15 | **27.09** | 7.28 |
| *w/o Thinking* | 12.98 | 11.76 | **18.03** | **11.30** | 48.76 | **2.41** | 30.60 | **2.03** | 26.90 | **6.76** |
| QWEN3-14B | **17.81** | 8.87 | 22.06 | 8.78 | 50.47 | 2.25 | 31.77 | 1.95 | **30.72** | 5.93 |
| *w/o Thinking* | 17.42 | **8.84** | **22.82** | **8.45** | **51.42** | **2.09** | **32.02** | **1.86** | 29.85 | **5.70** |
| QWEN3-32B | 17.33 | 9.54 | 23.70 | 8.41 | 51.01 | **1.96** | 31.84 | 1.66 | **31.68** | 5.80 |
| *w/o Thinking* | **18.13** | **8.39** | **24.10** | **7.58** | **51.59** | **1.96** | **32.18** | 1.71 | 30.48 | **5.59** |

Table 1: BLEU and MetricX scores for ten English → X directions from FLORES 200, with thinking (first line) and without thinking (second line). Best results are highlighted in bold.

## 4 APPROACHES TO IMPROVING MT WITH INTERMEDIATE REASONING

Given that general-purpose RMs do not seem to benefit from outputting thinking tokens prior to translation, we investigate how to build a successful "thinking" MT model, i.e., one that first produces intermediate reasoning before translation and outperforms models trained without intermediate steps. We apply COT FINE-TUNING (COTFT), whereby a student model is trained to first produce intermediate tokens (CoT or "thoughts") before generating the final target translation, as shown in Figure 1. Using a parallel dataset, we explore what types of intermediate information can be generated by a teacher model to train a "thinking" MT model (student). We explore multiple approaches (see the right side of Figure 1), which we categorize into two types:

- **CoT prompting.** This corresponds to the standard CoT distillation approach inherited from reasoning tasks (Figure 1, right, first box). For each source–target pair, the teacher is fed with a curated CoT prompt inspired by human translation strategies. It produces a reasoning trace explaining how to obtain the target from the source, or justifying why the given target is a correct translation of the source. In doing so, the model emulates the strategies used by human translators. It produces a first-person thought process in which it explains how it analyzes the sentence—identifying elements such as the subject, verb, and object—and how it arrives at the target translation by reasoning about linguistic aspects (syntactic rules, word order) and the broader context. Further details are provided in Appendix A.3.

- **Modular translation-specific prompting strategies.** Instead of adopting the classical approach, we propose using as intermediate information the traces obtained after applying modular translation-specific prompting strategies to translate the source. As mentioned in Section 2, they generally involve multiple steps, (see the five other boxes of Figure 1):
  - **MAPS**: a modular process comprising source analysis (extraction of keywords, topics, and relevant demonstrations) and corresponding translation attempts (each inspired by the extracted information), complemented by zero-shot translation.

- **SBYS**: a four-step process comprising pre-drafting research (identification of expressions that may pose a challenge for translation), drafting, refinement, and proofreading (for terminology, fluency, etc.).
- **TEaR**: a three-step process comprising translation (in a few-shot setting), annotation (of potential translation errors), and refinement (based on these annotations).
- **Self-Refine**: an iterative process comprising the initial translation (in a zero-shot setting) and successive rounds of self-refinement (to improve accuracy and fluency).
- **CompTra**: a three-step process comprising decomposition of the source into simpler phrases, few-shot translation of each phrase, and recombination[3] (into a final output).

Given a source sentence, we apply all steps of each selected prompting strategy and concatenate the outputs to form a text, which serves as intermediate information for CoTFT. We aim to see whether this approach results in improved MT models, analogous to how RMs improve over direct IO prompting with standard LLMs.

We compare CoTFT to INPUT-OUTPUT FINE-TUNING (IOFT), the baseline approach where the student is trained to directly predict the target translation given a source sentence. In both cases, the source and target are the same and the difference only lies in the presence of intermediate reasoning during training. We provide example training samples for each type of intermediate information in Appendix C.2. In summary, we first examine whether prompting a teacher to emulate a human translator and produce CoT traces helps to produce thinking MT model (student) that are more effective than a standard IOFT model. Finally, we evaluate whether MT-specific prompting strategies produce higher-quality traces for training thinking MT models and discuss the broader implications.

## 5 EXPERIMENTS

### 5.1 EXPERIMENTAL SETUP

**Evaluation Datasets.** Our main evaluation dataset is FLORES-200 (Goyal et al., 2022; Costa-jussà et al., 2022) devtest set (1012 examples). For fine-tuning (distillation experiments), we focus on two languages: Xhosa, an LRL, in the main paper, and Lithuanian, a HRL, in the appendix.

**Fine-tuning Datasets.** For Xhosa, we use `Llama-4-Scout-17B-16E-Instruct` (AI at Meta, 2025) and synthetic, multi-domain sentence-level data generated using the TOPXGEN pipeline (Zebaze et al., 2025b).[4,] For Lithuanian, we use the WMT19 dataset (Barrault et al., 2019) for training and run the same experiments as for Xhosa, as detailed in Appendices B.4 to B.6.

**Models.** For Xhosa, we use `Llama-4-Scout-17B-16E-Instruct` (AI at Meta, 2025) as the teacher (to generate reasoning traces for CoTFT) and `gemma-3-4b-pt` as the student. Ablation studies additionally consider `gemma-3-27b-it` (Gemma Team et al., 2025) and `DeepSeek-R1-Distill-Llama-70B` (DeepSeek-AI et al., 2025) as alternative teachers. For Lithuanian, we pair `gemma-3-27b-it` as the teacher with `gemma-3-1b-pt` as the student.

**Evaluation Metrics.** Our main evaluation metric is MetricX-24 (Juraska et al., 2024). We use the reference-based version `MetricX-24-Hybrid-XXL`, which supports the same 101 languages as mT5 (Xue et al., 2021). MetricX assigns a score ranging from 0 to 25, with higher scores indicating more errors in the translation. We also evaluate using BLEU[5] (Papineni et al., 2002) as implemented in sacreBLEU (Post, 2018).

**Implementation Details** We fine-tune all our models for 5k steps on one H100 80G with a learning rate of 1e-5, a constant scheduler with 500 warm-up steps (from 1e-6) and a batch size of 4. For IOFT we use 4 gradient accumulation steps and a maximum sequence length equal to 512,

---

[3]We do not use the output of the recombination step when building the intermediate tokens.

[4]The pipeline enables the generation of English–LRL parallel data (in this cases the LRL being Xhosa) from LLM-generated LRL texts, backtranslated into English as a way of alleviating the scarcity of diverse, high-quality datasets for LRLs).

[5]nrefs:1|case:mixed|eff:no|tok:flores200|smooth:exp|version:2.4.2

whereas for COTFT we use 16 gradient accumulation steps and a maximum sequence length of 2048. All models are evaluated in a zero-shot fashion with greedy decoding unless stated otherwise. See Appendix A.1 and A.2 for additional details and Appendix C.1 for ablation studies.

## 5.2 DISTILLED CHAIN-OF-THOUGHT AS INTERMEDIATE TOKENS

We compare IOFT and COTFT in the CoT distillation setup when the teacher is Llama-4-Scout-17B-16E-Instruct. We evaluate each of the six "CoT instance construction" prompt templates reflecting human translators' reasoning proposed by Feng et al. (2025a) for generating cold-start data for their R1-T1 model: Hierarchical translation, Triangulating translation, Backtranslation, Context-aware translation, translation explanation and structural transformation (see Appendix A.3). We fine-tune gemma-3-4b-pt and compare the performance of all six COTFT variants against IOFT. It is important to recall that in all scenarios, the source and target are the same, only presence or absence of traces and their template change. We report the BLEU and MetricX scores on FLORES-200 every 200 steps in Figure 3. We observe that COTFT consistently fails to improve over IOFT (in black) across all templates. The variability of performance across templates is negligible; they fall short compared to IOFT by about 0.5 BLEU and 0.5 MetricX points. We ran the same experiment with DeepSeek-R1-Distill-Llama-70B as the teacher and reached the same conclusions (see Appendix B.7).

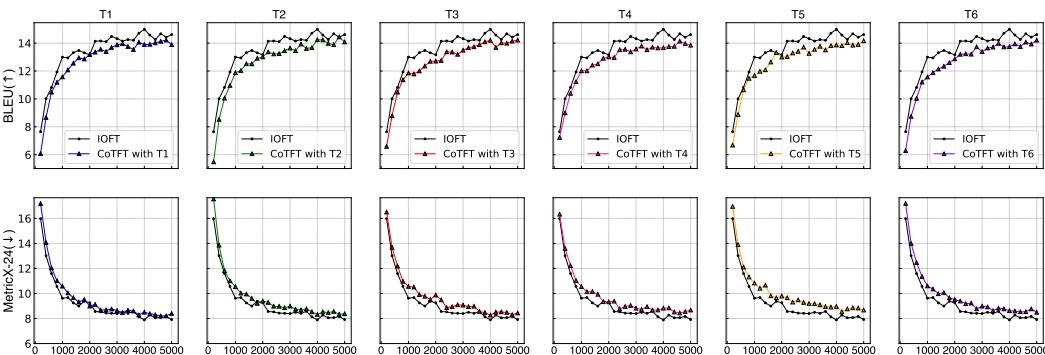

Figure 3: Comparison between IOFT and COTFT with six different CoT templates. Across all figures, the x-axis represents the number of training steps.

## 5.3 MT TRACES GENERATED BY PROMPTING STRATEGIES AS INTERMEDIATE TOKENS

RMs were built based on the premise that a "thinking process" formalized with natural language could help achieve better results. Asking an LLM to translate a sentence step-by-step does not improve over CoT-free zero-shot MT. However multistep prompting strategies that mimic translation reasoning exist. Given a teacher and a prompting strategy, can the traces generated during translation, when used as intermediate information, help produce better outputs? We consider five modular prompting strategies: MAPS, SBYS, TEaR, Self-Refine and CompTra. As shown in Figure 4, COTFT on traces based on such strategies outperforms IOFT, e.g. we get gains up to 3.5 BLEU and 2.0 MetricX with MAPS. For the other prompting strategies, improvements remain around +2 BLEU and -1.5 MetricX. Using CoT traces derived from these strategies appears beneficial—but why? The key difference is that, unlike pure CoT prompting, most of these strategies (except CompTra) include one or multiple drafting phases. The success of COTFT may therefore stem from drafts that surpass the ground truth. We test this hypothesis as follows. For each strategy, we use the quality estimation score BLASER 2.0-QE (Duquenne et al., 2023; Dale & Costa-jussà, 2024) to obtain the best translation between the ground truth and the attempts embedded in the teacher's traces. We consider 2 scenarios. IOFT-MAX(STRATEGY) which is IOFT where the target is replaced by the best one between the ground truth and those potentially generated by the prompting strategy. COTFT-MAX(STRATEGY) which is analogous to IOFT-MAX(STRATEGY) but with the intermediate tokens. In addition to the above scenarios, we consider IOFT-BOA (best of all) which is IOFT where the target is the best between the ground truth and translations embedded into the traces obtained across all prompting strategies considered (MAPS, SBYS, TEaR, Self-Refine and CompTra).

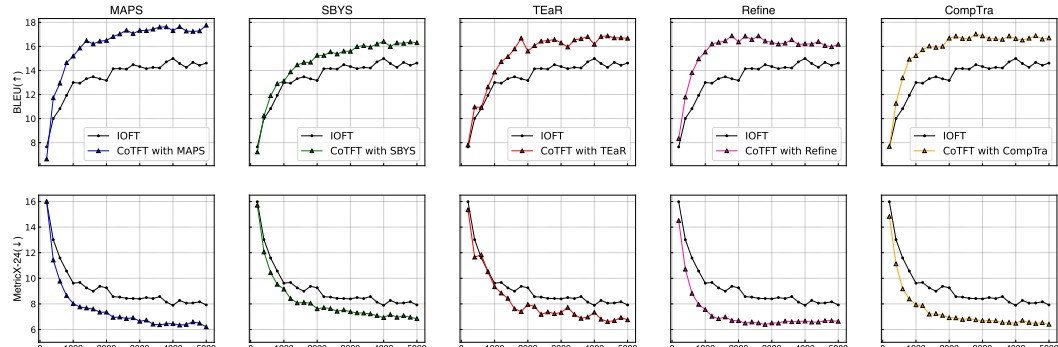

Figure 4: Comparison between IOFT and CoTFT with five different prompting strategies.

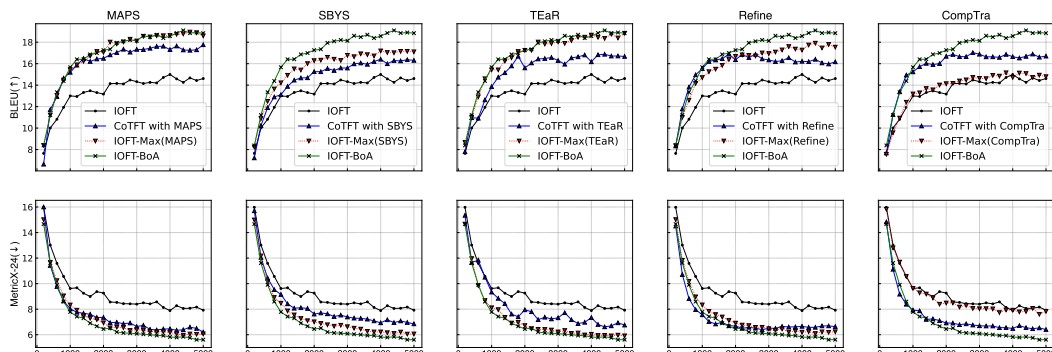

Figure 5: Comparison between IOFT and CoTFT with five different prompting strategies.

**First scenario.** For MAPS, SBYS, TEaR and Self-Refine, IOFT-MAX (in red) works better than CoTFT (i.e. with the traces) and IOFT (Figure 5). This indicates that the quality of the target is an important factor for downstream performance. Using better ground truths (IOFT-BOA, in green) can make IOFT go from 14 BLEU to 18 BLEU (8 MetricX to 5.6 MetricX) with the same number of parallel pairs and the same training recipe. Interestingly, CompTra behaves differently. As a matter of fact, the traces of CompTra only contain translations of small sentences built by splitting the source, not of the source itself. The translations of these small phrases are unlikely to be better than the ground truth. This explains the performance similarity between standard IOFT and IOFT-MAX(COMPTRA). CoTFT with CompTra outperforms IOFT-MAX(COMPTRA) and IOFT indicating that CoTFT can be successful without including better translation attempts than the ground truth; partial translations are enough.

**Second Scenario.** For MAPS, SBYS, TEaR and Self-Refine, IOFT-MAX generally works better than CoTFT-MAX (Figure 6). This confirms the previous conclusion on these strategies, i.e. when the traces provided by the teacher do not contain translation attempts better than the ground truth, they do not help improve the MT performance. CoTFT-MAX(COMPTRA) works slightly better than IOFT-MAX(COMPTRA), but both underperform CoTFT with CompTra. This reinforces the idea that sentence-translation pairs (related to the sentence considered but smaller and different) can serve as valuable intermediate information for CoTFT.

IOFT-BOA (in green) being consistently above all the curves suggest that the quality of the target translations matters more than traces, and IOFT with better ground truths outperforms CoTFT while being cheaper and faster to train. We obtained the exact same results when we use `gemma-3-27b-it` as the teacher in Appendix B.8. Likewise, Appendix B.10 shows that using `gemma-3-12b-pt` as the student leads to identical conclusions.

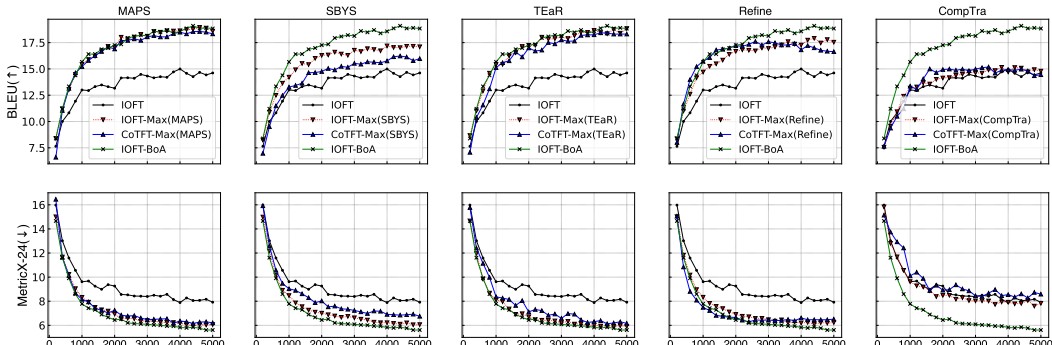

Figure 6: Comparison between IOFT and CoTFT with five different prompting strategies.

# 6 DISCUSSION AND ANALYSIS

## 6.1 DOWN THE RABBIT HOLE OF SENTENCE DECOMPOSITION

We further investigate the generation of sentence-translation pairs as intermediate tokens. With CompTra, the pairs are obtained by decomposing the source into multiple phrases (Zebaze et al., 2025a). We consider three other decomposition strategies: *Paraphrases (P)*, *Syntactic Paraphrases (SP)* and *Hard Expressions (H)*. *S* asks the teacher to generate five paraphrases of the source. *SP* generates five sentences with the same syntax as the source (grammatical roles, syntactic dependencies etc.) but using different words. Finally, *H* asks the teacher to extract words or expressions it deems difficult to translate. In all cases, the teacher translates the expressions generated after decomposition. For each decomposition strategy (*P*, *SP*, *H*, and CompTra), we compare CoTFT (which uses the teacher's sentence–translation pairs as reasoning traces) with IOFT. We also evaluate IOFT-EXT(*strategy*), which applies IOFT on the original dataset augmented with the generated pairs as additional training samples.

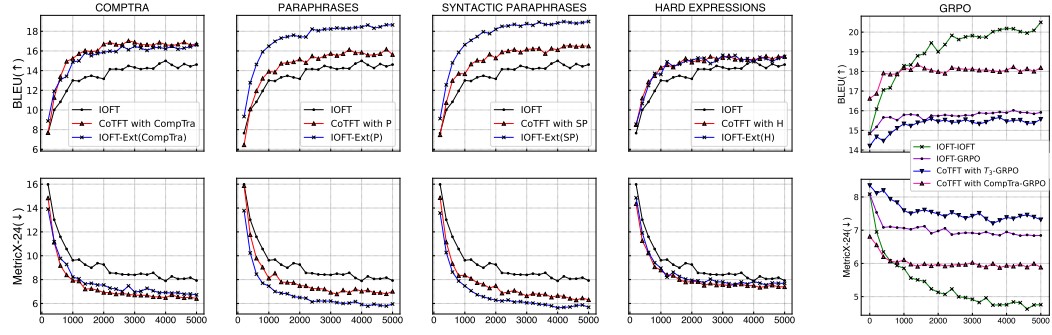

Figure 7: Comparison between IOFT and CoTFT with four sentence decomposition strategies (left) and with GRPO (right).

As shown in Figure 7 (8 leftmost panels), CoTFT consistently outperforms IOFT across all decomposition strategies. CompTra and *SP* are the best approaches with CoTFT. IOFT-EXT(P) and IOFT-EXT(SP) result in significant gains over the IOFT baseline (+4 BLEU, -2 MetricX). IOFT-EXT also outperforms CoTFT with *P* and *SP*. However, it has less success with *H* and CompTra. We attribute this to the fact that the pairs generated by *H* and CompTra are shorter and largely overlap with the original training samples, giving fewer gains as additional data compared to entirely new sentences. However, these short phrases (generated via CompTra or *H*) are valuable intermediate information, as CoTFT outperforms IOFT and IOFT-EXT in these scenarios. IOFT-EXT(SP) and IOFT-EXT(P) are the best overall, showing the large impact of the amount of parallel data.

## 6.2 REINFORCEMENT LEARNING AFTER IOFT AND COTFT

Finally, we investigate whether CoTFT improves performance during RL fine-tuning. We consider three setups: IOFT, CoTFT with CompTra and CoTFT with $T_3$ (like in Section 5.2). The final checkpoints (checkpoint-5000) are further fine-tuned with GRPO (Shao et al., 2024) on a second parallel dataset[6] (on 3 GPUs for 5000 steps, more details in Appendix A.5). We consider three reward functions: one based on the BLEU and chrF++ scores with the ground truth, a second using COMET-22 (`wmt22-cometkiwi-da`; Rei et al., 2022) and a last one based on the BLASER-2.0 QE scores between the sources and hypotheses. For CoTFT, we consider an additional format reward to ensure that the models preserve their prior thinking before translating. We compare all three RL fine-tunings with IOFT on the second dataset (on 1 GPU for 5000 steps) and report the results in Figure 7 (two rightmost panels). The ordering after SFT (CoTFT with $T_3 \leq$ IOFT $\leq$ CoTFT with CompTra) remains unchanged after RL, with gains of about +1.3 BLEU and -1.0 MetricX in all setups. Notably, CoTFT still does not outperform IOFT, even with RL. This is consistent with Zheng et al.'s (2025) findings, namely that CoT signals fail to induce meaningful reasoning when the reward is applied only to the final translation. Moreover, unlike mathematics where step-by-step explanations are widely present in pre-training corpora (proofs), it is not the case for translation data. This scarcity of reasoning-like data may explain CoT's limited effectiveness in MT. Moreover, in the context of RL for MT, the notion of a "verifiable reward" is not well-defined. The idea of a "correct translation" is far less absolute than a correct answer in mathematics (e.g. 1+1=2). As a result, we rely on proxy metrics such as BLEU or COMET, but using these rewards do not reproduce the dramatic improvements observed for mathematics and code. Finally, we find that continuing SFT (IOFT) on the IOFT checkpoint gives much larger gains (+6 BLEU, -3 MetricX) than GRPO which quickly stagnates, reinforcing that standard IOFT alone can achieve superior MT performance. We report the results of applying GRPO on checkpoints obtained after CoTFT with MAPS, SBYS, TEaR and Self-Refine in Appendix B.9.

## 7 CONCLUSION

We explored fine-tuning LLMs to generate intermediate tokens as a method to improve their MT capabilities. Through a broad spectrum of experiments, we find that outputting reasoning traces does not help models to produce better translations (for thinking models and during CoT distillation). We also investigated how traces produced by alternative MT prompting strategies could help and found that parallel pairs can serve as valuable intermediate information. However, ultimately two factors affect the success of MT fine-tuning: the quality of the target translation and the quantity of parallel data. When they are both ensured, standard IOFT goes a long way. These findings generalize two important results in MT: (i) The inability of CoT prompting to improve over standard IO prompting in zero-shot with standard LLMs (Peng et al., 2023; Zebaze et al., 2025a; Nguyen & Xu, 2025), and (ii) the success of approaches using external resources such as grammars comes from the presence of parallel sentences in the grammar (Aycock et al., 2025; Marmonier et al., 2025). CoT (intermediate tokens) provided no benefit when translation attempts (full or partial) were absent, but accounted for all improvements when they were present. MT behaves differently from typical reasoning tasks. Unlike base LMs, for which CoT prompting considerably degrades MT performance in comparison to IO prompting, the "thinking mode" in TMs does not degrade performance, but it also fails to give improvements (Section 3). This likely stems from the fact that TMs are explicitly trained to think, whereas standard LMs often produce "thinking tokens" that are not helpful or deleterious for MT. It is possible that the limited effectiveness of TMs in MT comes from their thinking-oriented training not containing enough real-world examples of "thinking in the context of MT." As discussed before, we attribute this largely to the scarcity of such traces and their weak relevance compared to the final translation. Section 5.2 further shows that generating reasoning-like traces for MT using LLMs produces intermediate steps that do not enable CoTFT to surpass IOFT. In contrast, partial or full translation attempts appear to provide the most useful intermediate signals—mirroring the behaviour of MT prompting strategies, which often rely on producing multiple translation attempts before generating the final answer. Ultimately, data quantity and quality win out: a polished translation dataset without additional reasoning tokens goes a long way.

---

[6]Based on `https://hf.co/datasets/almanach/topxgen-gemma-3-27b-and-nllb-3.3b`

ETHICS STATEMENT

This paper presents work whose goal is to advance the field of Machine Translation with Large Language Models. There are many potential societal consequences of our work, none which we feel must be specifically highlighted here. We used language models in writing but only for polishing and proofreading.

REPRODUCIBILITY STATEMENT

For this paper, we have worked with open-source models which are available on the Hugging Face Hub. Our datasets are already available or will be released as artifacts of this work. We have already included our training and inference hyperparameters as well as prompts in Section 5.1 and Appendix A to foster reproducibility. Finally, we also release our code for data generation, fine-tuning and evaluation as supplemental material.

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

## A REPRODUCIBILITY DETAILS

### A.1 MODELS, DATASETS AND TOOLS

In Table 2, we list the links to the relevant resources used for the experiments.

### A.2 IMPLEMENTATION DETAILS

We use HuggingFace's Transformers library (Wolf et al., 2020; Gugger et al., 2022). We adopt the prompt template introduced by Xu et al. (2024a), and compute the loss only on the target (translation or intermediate tokens followed by the translation). We use the same prompt when evaluating the checkpoints. During CoTFT the target is formatted as `<think>\n{Intermediate Tokens}\n</think>\n\nFinal Translation\n{Target translation}`.

```
Translate this from English to Hausa:
English: "We now have 4-month-old mice that are non-diabetic that used to
    be diabetic," he added.
Hausa:
```

For instruction-following models and thinking models we use the following evaluation prompt.

```
Please write a high-quality Xhosa translation of the following English
    sentence

"We now have 4-month-old mice that are non-diabetic that used to be
    diabetic," he added.

Please provide only the translation, nothing more.
```

| Datasets | |
|---|---|
| FLORES-200 | https://huggingface.co/datasets/facebook/flores |
| NTREX HF | hhttps://huggingface.co/datasets/mteb/NTREX |
| TICO-19 | https://huggingface.co/datasets/gmnlp/tico19 |
| Models evaluated | |
| Qwen3-0.6B | https://huggingface.co/Qwen/Qwen3-0.6B |
| Qwen3-1.7B | https://huggingface.co/Qwen/Qwen3-1.7B |
| Qwen3-4B | https://huggingface.co/Qwen/Qwen3-4B |
| Qwen3-8B | https://huggingface.co/Qwen/Qwen3-8B |
| Qwen3-14B | https://huggingface.co/Qwen/Qwen3-14B |
| Qwen3-32B | https://huggingface.co/Qwen/Qwen3-32B |
| DeepSeek-R1-Distill-Qwen-1.5B | https://huggingface.co/deepseek-ai/DeepSeek-R1-Distill-Qwen-1.5B |
| DeepSeek-R1-Distill-Qwen-7B | https://huggingface.co/deepseek-ai/DeepSeek-R1-Distill-Qwen-7B |
| DeepSeek-R1-Distill-Llama-8B | https://huggingface.co/deepseek-ai/DeepSeek-R1-Distill-Llama-8B |
| DeepSeek-R1-Distill-Qwen-14B | https://huggingface.co/deepseek-ai/DeepSeek-R1-Distill-Qwen-14B |
| DeepSeek-R1-Distill-Qwen-32B | https://huggingface.co/deepseek-ai/DeepSeek-R1-Distill-Qwen-32B |
| DeepSeek-R1-Distill-Llama-70B | https://huggingface.co/deepseek-ai/DeepSeek-R1-Distill-Llama-70B |
| Gemma-3-27B-It | https://huggingface.co/google/gemma-3-27b-it |
| Gemma-3-4B-Pt | https://huggingface.co/google/gemma-3-4b-pt |
| Gemma-3-1B-Pt | https://huggingface.co/google/gemma-3-1b-pt |
| Llama-4-Scout-17B-16E-Instruct | https://huggingface.co/meta-llama/Llama-4-Scout-17B-16E-Instruct |
| Other resources | |
| MetricX24-Hybrid-XXL | https://huggingface.co/google/metricx-24-hybrid-xxl-v2p6 |
| XCOMET-XXL | https://huggingface.co/Unbabel/XCOMET-XXL |
| FastText | https://huggingface.co/facebook/fasttext-language-identification |
| vLLM (Kwon et al., 2023) | https://github.com/vllm-project/vllm |

Table 2: Links to datasets, benchmarks and models.

We calculate statistical significance using bootstrap resampling (Koehn, 2004) with 300 samples of 500 sentences and a $p$-value threshold of 0.05.

## A.3 CoT CONSTRUCTION TEMPLATES

We use the six CoT construction templates proposed by He et al. (2025). They mimic reasoning strategies for translation commonly adopted by human translators.

T1 is **Hierarchical Translation**

```
<think>
1. Analyze the sentence structure and identify the core elements (subject
   , verb, object).
2. Translate the sentence from the origin language to the target language
   , focusing on the core elements.
3. Review the translation for basic accuracy and grammatical structure.
4. Identify areas that need further refinement (e.g., word choice, tense,
    or word order).
5. Modify the translation to improve fluency and coherence, considering
   the context.
6. Finalize the translation by ensuring it retains the original meaning
   while improving readability.
</think>
```

T2 is **Triangulating Translation**

```
<think>
1. Identify basic elements: Break down the sentence into its main
   components and identify the key subject, verb, and object.
2. Translate to intermediate language: Convert these elements into an
   intermediate language structure (e.g., simple syntactic rules or
   function names).
3. Refine back to target language: Translate from the intermediate
   language back to the target language, adjusting for syntactic norms
   and idiomatic expressions.
4. Check for accuracy: Ensure that the meaning is preserved in the
   translated sentence by checking noun-verb agreement and connectors.
5. Adjust word order: Modify word order to ensure that it aligns with the
    target language's grammatical structure.
```

```
6. Final refinement: Review the translation for naturalness, idiomatic
   use, and overall flow.
</think>
```

### T3 is **Back Translation**

```
<think>
1. Analyze the provided context in the source language.
2. Translate the source text to the target language.
3. Perform back translation from the target language to the source
    language.
4. Compare the back translation with the original source context.
5. Evaluate whether the meaning of the back translation aligns with the
    original.
6. If discrepancies are identified, adjust the target language
   translation to enhance consistency with the original meaning.
7. Finalize the translation by ensuring both forward and back
   translations accurately align across all languages involved.
</think>
```

### T4 is **Context-aware Translation**

```
<think>
1. Analyze the current sentence, along with the previous sentences, to
   understand the overall conversation context.
2. Identify key elements like tone, formality, or subject matter based on
    the ongoing conversation.
3. Translate the sentence while ensuring that the translation is aligned
   with the tone, style, and subject of the preceding dialogue.
4. If any ambiguity exists in the translation due to context, refine the
   translation to better fit the conversation flow.
5. Verify that the translation maintains coherence with the larger
   conversation, ensuring consistency in language and tone.
6. Finalize the translation by cross-checking it with the conversation's
   context to ensure it feels natural and appropriately aligned.
</think>
```

### T5 is **Translation Explanation**

```
<think>
1. Analyze the source sentence and identify the key elements (verbs,
    subjects, objects, etc.).
2. Based on these elements, determine the most suitable translation
   strategy (literal vs. idiomatic).
3. Select the best translation for each word or phrase, considering
   context and languagespecific structures.
4. Explain the rationale behind choosing specific words or phrases.
5. After completing the initial translation, review each translation
   decision and explain any adjustments made for fluency or accuracy.
6. Provide a final explanation for the translation choices, discussing
    any trade-offs made between literal meaning and contextual
    appropriateness.
</think>
```

### T6 is **Structural Transformation**

```
<think>
1. Analyze the sentence's syntactic structure in the source language (e.g
    ., identify whether it's active or passive).
2. Determine the most appropriate syntactic structure in the target
   language (e.g., whether it needs to be rephrased from active to
   passive or vice versa).
3. Adjust the word order and grammatical structure in the target language
    to match the sentence's meaning, while maintaining clarity.
```

```
4. Translate the sentence, ensuring that subject-verb-object
     relationships and other syntactic elements align with target language
     norms.
5. After the translation, check the sentence's grammar and overall flow
     in the target language, making sure it is clear and fluid.
6. If the sentence feels awkward or unnatural, refine the structure by
     adjusting word choice or reordering components.
</think>
```

These approaches serve as more structured alternatives to the basic T0 strategy, where the teacher is simply prompted to choose whatever translation procedure it deems most appropriate for the input.

```
Explain step by step how to translate the source sentence into the target
     sentence.
```

Given a CoT template, we use the following prompt to obtain a CoT produced by a teacher explaining how to obtain the provided translation of a given source sentence following the strategy corresponding to the template. The CoT produced is in the first-person and can later be used for CoTFT.

```
Assume that you are a student engaged in translating a sentence from {src
     } to {tgt}.
Now you have both the source sentence and the target sentence, and need
     to analyze how to translate
from the source sentence to the given target sentence based on the
     provided Thinking Chain Guide. And
output the chain-of-thought trajectory from source to target sentence.

The {src} statement is as follows:
<Source Sentence>
{sentence}
</Source Sentence>

The {tgt} statement is as follows:
<Target Sentence>
{translation}
</Target Sentence>

You continuously reflect on how to translate the source sentence to the
     given target sentence
based on the thinking guidance provided.

The given Thinking Chain Guide is as follows:
<Thinking Chain Guide>
{chain_of_thought_template}
</Thinking Chain Guide>

Please refine the entire analysis process into a complete self-reflective
     description (in the present tense). For self-reflection, you can
     refer to the following thinking steps:
directly output the self-reflective description in the <think></think>
     tags, without any additional descriptions or explanations.
Each line in the reflective description can be viewed as a reasoning step
     in the translation process.
```

## A.4 PROMPTING STRATEGIES

**Step-by-Step Translation (SBYS)**:

```
{predrafting research}

{draft translation}
```

```
Now let's move to the next stage: Post-editing with local refinement.
In this stage, the primary aim is to refine the draft translation by
    making micro-level improvements that improve the draft's fluency.

Here is a refined version of the translation
{refinement}

Now, we will proofread the refined text for grammar spelling, punctuation
    , terminology and overall fluency."

Here is the translation after proofreading
{proofreading}

We will further improve it to obtain the final, polished translation.
```

**Multi-Aspect Prompting and Selection (MAPS):**

```
Here is a draft translation

1. {zero-shot translation}

Let's write an English sentence related to but different from the input
    English sentence and translate it into {language}

{demonstrations}

Given this knowledge, we can draft another translation

2. {demonstrations-inspired translation}

Let's extract the keywords in the provided English sentence, and then
    translate these keywords into {language}

{keywords}

Given this knowledge, we can draft another translation

3. {keywords-inspired translation}

Let's use a few words to describe the topics of the provided English
    sentence

{topics}

Given this knowledge, we can draft another translation

4. {topics-inspired translation}

We will choose the best of these translations and further improve it to
    obtain the final, polished translation.
```

**Self-Refine**

```
Here is a draft translation

1. {draft translation}

Let's improve it and write a better translation

2. {refinement 1}

Let's further improve it and write a better translation

3. {refinement 2}
```

```
Let's improve it one last time and write a better translation

4. {refinement 3}

We will choose the best of these translations and further improve it to
    obtain the final, polished translation.
```

**Translate, Estimate and Refine (TEaR)**

```
Here is a draft translation

1. {draft translation}

Let's identify errors and assess the quality of the draft translation.
The categories of errors are accuracy (addition, mistranslation, omission
    , untranslated text), fluency (character encoding, grammar,
    inconsistency, punctuation, register, spelling), locale convention (
    currency, date, name, telephone, or time format) style (awkward),
    terminology (inappropriate for context, inconsistent use), non
    translation, other, or no-error.
Each error is classified as one of three categories: critical, major, and
     minor. Critical errors inhibit comprehension of the text. Major
    errors disrupt the flow, but what the text is trying to say is still
    understandable. Minor errors are technical errors but do not disrupt
    the flow or hinder comprehension.

Here are the MQM annotations of the draft:
{MQM annotations}

Upon reviewing the translation and error information, we can refine the
    draft and obtain a better translation

2. {refinement}

We will further improve it to obtain the final, polished translation."
```

**Compositional Translation (CompTra)**

```
1. English Sentence
{}
Xhosa Translation
{}

2. English Sentence
{}
Xhosa Translation
{}

3. English Sentence
{}
Xhosa Translation
{}
```

### A.5 RL TRAINING HYPERPARAMETERS

For GRPO, we use the Hugging Face TRL library (von Werra et al., 2020). Training is conducted on four H100 GPUs, with one dedicated to model deployment for reward computation. We set a per-device batch size of 4 with 4 gradient accumulation steps, for a total of 5000 steps including 100 warmup steps. Hyperparameters include a beta value of 0.02, a maximum gradient norm of 1.0, and a temperature of 1.0. For generation, we sample 12 outputs per prompt with an effective batch size

of 48. We apply LoRA (Hu et al., 2022), fine-tuning the `q_proj`, `k_proj`, `v_proj`, `o_proj`, `gate_proj`, `up_proj`, and `down_proj` modules with rank $r = 32$, scaling factor $\alpha = 64$, and dropout rate 0.05.

# B  ADDITIONAL EXPERIMENTS

| Models | Czech | | Finnish | | French | | German | | Japanese | |
|---|---|---|---|---|---|---|---|---|---|---|
| | chrF++ | XCOMET | chrF++ | XCOMET | chrF++ | XCOMET | chrF++ | XCOMET | chrF++ | XCOMET |
| QWEN3-0.6B | **19.43** | 16.52 | 16.36 | 13.56 | **47.19** | 53.77 | **39.34** | 71.05 | **14.11** | 44.66 |
| *w/o Thinking* | 18.53 | **55.22** | **18.12** | **23.62** | 45.94 | **54.82** | 37.75 | **74.96** | 11.35 | **53.73** |
| QWEN3-1.7B | 33.89 | **39.91** | 26.45 | 20.00 | 57.40 | 80.58 | 49.52 | 88.42 | **21.39** | **69.82** |
| *w/o Thinking* | **35.27** | 38.62 | **29.54** | **22.68** | **57.68** | **80.67** | **49.89** | **89.58** | 20.81 | 67.83 |
| QWEN3-4B | 43.43 | **68.22** | 36.99 | **45.99** | **62.49** | **89.14** | **55.13** | 93.48 | 24.68 | **80.26** |
| *w/o Thinking* | **43.92** | 65.45 | **38.11** | 42.54 | 62.26 | 88.23 | 55.07 | **94.25** | **26.53** | 79.23 |
| QWEN3-8B | 48.49 | **80.32** | 43.15 | **65.78** | 64.68 | **92.19** | **58.30** | 95.94 | 27.24 | 86.18 |
| *w/o Thinking* | **48.70** | 77.64 | **43.44** | 62.60 | **65.16** | 91.89 | 58.21 | **96.38** | **28.50** | **86.60** |
| QWEN3-14B | **51.57** | **86.91** | 46.26 | **76.27** | 66.41 | 92.99 | **60.52** | 96.89 | 29.88 | **89.88** |
| *w/o Thinking* | 51.29 | 84.65 | **46.78** | 74.75 | **66.91** | **93.57** | 60.18 | **97.37** | **30.54** | 89.76 |
| QWEN3-32B | **51.45** | **86.59** | 46.47 | **77.18** | **65.99** | 92.79 | 60.35 | 96.27 | 29.10 | 89.38 |
| *w/o Thinking* | 50.85 | 86.48 | **47.40** | **77.13** | 65.98 | **93.81** | **60.38** | **97.60** | **30.89** | **89.97** |

| Models | Kazakh | | Lithuanian | | Portuguese | | Spanish | | Turkish | |
|---|---|---|---|---|---|---|---|---|---|---|
| | chrF++ | XCOMET | chrF++ | XCOMET | chrF++ | XCOMET | chrF++ | XCOMET | chrF++ | XCOMET |
| QWEN3-0.6B | 4.14 | 15.77 | 10.73 | 14.19 | **48.05** | 67.21 | **41.70** | 70.45 | 22.64 | 18.39 |
| *w/o Thinking* | **6.78** | **20.31** | **12.97** | **16.91** | 44.73 | **68.36** | 40.29 | **72.40** | **23.77** | **22.97** |
| QWEN3-1.7B | 7.51 | 15.09 | 24.83 | 18.80 | **58.97** | **87.43** | 48.25 | **87.84** | 34.08 | 39.47 |
| *w/o Thinking* | **10.84** | **15.62** | **26.11** | **19.52** | 58.99 | 87.07 | **48.61** | 87.55 | **35.89** | **43.02** |
| QWEN3-4B | **26.54** | **25.20** | 36.26 | **43.79** | **63.93** | **92.69** | 50.66 | **92.23** | 42.97 | **66.46** |
| *w/o Thinking* | 26.51 | 23.78 | **36.60** | 42.88 | 63.70 | 92.17 | **51.07** | 92.20 | **43.52** | 65.43 |
| QWEN3-8B | **34.64** | **40.48** | 41.54 | **61.88** | 65.66 | 94.16 | 52.45 | 94.13 | **47.86** | **77.71** |
| *w/o Thinking* | 33.86 | 36.52 | **42.02** | 60.84 | **65.77** | **94.60** | **52.55** | **94.62** | 47.90 | 77.68 |
| QWEN3-14B | **39.53** | **53.44** | 45.08 | **73.81** | 66.70 | 95.12 | 53.29 | 94.64 | **50.97** | **83.68** |
| *w/o Thinking* | 38.95 | 50.77 | **45.82** | 73.72 | **67.55** | **95.31** | **53.60** | **95.25** | 50.42 | 83.35 |
| QWEN3-32B | 38.34 | **53.66** | 45.92 | 75.60 | 66.71 | 94.93 | 53.19 | 95.07 | **51.29** | **84.03** |
| *w/o Thinking* | **39.22** | 52.05 | **46.68** | **76.89** | **67.64** | **95.77** | **53.57** | **95.32** | 50.60 | 83.08 |

Table 3: chrF++ and XCOMET scores for 10 English → X directions from FLORES 200. Best results are highlighted in bold.

| Models | Czech | | Finnish | | French | | German | | Japanese | |
|---|---|---|---|---|---|---|---|---|---|---|
| | BLEU | MetricX | BLEU | MetricX | BLEU | MetricX | BLEU | MetricX | BLEU | MetricX |
| DEEPSEEK-R1-DISTILL-QWEN-14B | 22.78 | 9.91 | **10.55** | 17.40 | **46.13** | 2.87 | **34.54** | 2.49 | 21.04 | 4.85 |
| *w/o Thinking* | **23.04** | **9.50** | 10.00 | **17.13** | 45.42 | **2.54** | 30.26 | **2.40** | **22.04** | **4.45** |
| DEEPSEEK-R1-DISTILL-QWEN-32B | 28.49 | 7.12 | 15.61 | 13.06 | 48.68 | 2.78 | 38.25 | 1.82 | 24.11 | 4.56 |
| *w/o Thinking* | **29.61** | **6.22** | **16.12** | **12.52** | **49.35** | **2.16** | **38.95** | **1.44** | **26.57** | **3.89** |
| DEEPSEEK-R1-DISTILL-LLAMA-70B | 37.31 | 4.19 | 29.90 | 5.35 | **52.34** | 2.22 | 43.39 | 1.16 | 25.61 | 4.08 |
| *w/o Thinking* | **38.47** | **3.52** | **30.96** | **4.38** | **52.34** | **1.88** | **44.78** | **0.91** | **27.94** | **3.62** |

| Models | Kazakh | | Lithuanian | | Portuguese | | Spanish | | Turkish | |
|---|---|---|---|---|---|---|---|---|---|---|
| | BLEU | MetricX | BLEU | MetricX | BLEU | MetricX | BLEU | MetricX | BLEU | MetricX |
| DEEPSEEK-R1-DISTILL-QWEN-14B | 2.38 | 13.25 | **7.87** | 20.02 | 46.28 | 2.86 | 29.48 | 2.56 | **17.99** | 12.79 |
| *w/o Thinking* | **2.50** | **6.49** | 7.21 | **18.79** | **47.11** | **2.52** | **30.52** | **2.13** | 17.78 | **11.70** |
| DEEPSEEK-R1-DISTILL-QWEN-32B | **5.66** | 18.37 | 13.24 | 16.09 | 49.17 | 2.55 | 30.62 | 2.26 | 24.52 | 9.36 |
| *w/o Thinking* | 4.95 | **16.19** | **14.02** | **14.92** | **50.53** | **2.02** | **31.67** | **1.74** | **24.96** | **7.88** |
| DEEPSEEK-R1-DISTILL-LLAMA-70B | 21.39 | 8.14 | 25.63 | 8.22 | 52.27 | 2.09 | 32.68 | 1.78 | 33.42 | 5.46 |
| *w/o Thinking* | **21.56** | **6.76** | **27.01** | **7.33** | **52.64** | **1.75** | **33.54** | **1.55** | **34.77** | **5.00** |

Table 4: BLEU and MetricX scores for 10 English → X directions from FLORES 200. Best results are highlighted in bold.

| Models | Czech | | Finnish | | French | | German | | Japanese | |
|---|---|---|---|---|---|---|---|---|---|---|
| | BLEU | MetricX | BLEU | MetricX | BLEU | MetricX | BLEU | MetricX | BLEU | MetricX |
| QWEN3-0.6B | **5.65** | 22.53 | **3.21** | **23.13** | **24.49** | **9.51** | **16.11** | **10.15** | **8.08** | **9.29** |
| *w/o Thinking* | 5.31 | **22.49** | 2.69 | 23.40 | 21.48 | 10.28 | 14.51 | 10.43 | 6.09 | 11.84 |
| QWEN3-1.7B | 15.92 | 14.68 | 7.86 | 19.18 | 38.28 | 4.69 | **28.41** | 4.30 | **16.46** | **5.85** |
| *w/o Thinking* | **16.26** | **14.48** | **8.50** | **18.71** | **38.52** | **4.54** | 28.19 | **4.05** | 15.29 | 5.89 |
| QWEN3-4B | 24.43 | 9.18 | 13.96 | 13.88 | 44.47 | 3.66 | 34.20 | 3.14 | 20.68 | 5.22 |
| *w/o Thinking* | **25.77** | **8.74** | **15.33** | **13.39** | **44.99** | **3.25** | **34.79** | **2.54** | **21.36** | **4.79** |
| QWEN3-8B | 30.11 | 6.65 | 19.37 | 10.21 | 48.89 | 2.89 | 38.88 | 1.92 | **24.88** | 4.28 |
| *w/o Thinking* | **30.36** | **6.59** | **19.70** | **9.89** | **49.18** | **2.58** | **39.16** | **1.59** | 24.67 | **4.10** |
| QWEN3-14B | **34.44** | 5.11 | 23.51 | 7.59 | 51.23 | 2.37 | **41.83** | 1.39 | 27.93 | 3.86 |
| *w/o Thinking* | 34.01 | **4.93** | **23.89** | **7.43** | **51.45** | **2.13** | 41.76 | **1.17** | **28.00** | **3.63** |
| QWEN3-32B | 15.69 | 15.01 | 11.19 | 16.04 | 29.14 | 11.67 | 22.99 | 11.00 | 16.14 | 12.00 |
| *w/o Thinking* | **34.34** | **4.64** | **24.91** | **6.72** | **50.35** | **1.99** | **42.58** | **1.09** | **27.86** | **3.53** |

| Models | Kazakh | | Lithuanian | | Portuguese | | Spanish | | Turkish | |
|---|---|---|---|---|---|---|---|---|---|---|
| | BLEU | MetricX | BLEU | MetricX | BLEU | MetricX | BLEU | MetricX | BLEU | MetricX |
| QWEN3-0.6B | 0.93 | **23.89** | **2.21** | **24.04** | **26.00** | **9.44** | **17.51** | **8.38** | **6.46** | 21.24 |
| *w/o Thinking* | **1.31** | 23.85 | 2.11 | 24.12 | 21.99 | 11.08 | 15.81 | 9.11 | 5.43 | **21.16** |
| QWEN3-1.7B | 2.03 | 22.03 | 6.80 | **20.51** | **39.27** | 4.41 | **25.76** | 3.85 | 14.10 | 14.36 |
| *w/o Thinking* | **3.24** | **21.58** | **7.69** | 20.65 | 39.22 | **4.23** | 25.63 | **3.76** | **14.71** | **13.41** |
| QWEN3-4B | 10.1 | 15.26 | 13.64 | 15.10 | 45.61 | 3.44 | 28.43 | 3.14 | 21.32 | 10.28 |
| *w/o Thinking* | **11.0** | **14.50** | **14.62** | **14.44** | **45.93** | **3.16** | **29.12** | **2.69** | **21.95** | **9.63** |
| QWEN3-8B | 15.64 | 11.07 | 19.01 | 10.97 | 49.03 | 2.64 | 30.87 | 2.30 | **27.01** | 7.34 |
| *w/o Thinking* | **15.77** | **10.80** | **19.51** | **10.67** | **49.44** | **2.47** | **31.19** | **2.09** | 26.98 | **6.82** |
| QWEN3-14B | **20.35** | 8.30 | 23.68 | 8.41 | 50.93 | 2.28 | 32.38 | 1.94 | **30.99** | 6.08 |
| *w/o Thinking* | 19.40 | **7.97** | **24.59** | **7.86** | **51.37** | **2.10** | **32.67** | **1.72** | 29.97 | **5.66** |
| QWEN3-32B | 10.85 | 15.88 | 11.89 | 16.26 | 29.30 | 11.61 | 17.69 | 11.53 | 16.08 | 14.69 |
| *w/o Thinking* | **20.63** | **7.61** | **25.28** | **7.12** | **50.89** | **1.95** | **33.05** | **1.58** | **30.97** | **5.26** |

Table 5: 5-shot BLEU and MetricX scores for 10 English → X directions from FLORES 200. Best results are highlighted in bold.

## B.1 BENCHMARKING LRMS AT SCALE: TO THINK, OR NOT TO THINK?

In this section we further investigate the impact of thinking tokens when benchmarking LRMs. In Table 3 we report the chrF++[7] (Popović, 2015; Popović, 2017) and XCOMET-XXL (Guerreiro et al., 2024) scores in the same setup as Table 1. They tell the same story as BLEU and MetricX. Outputting thinking tokens only marginally helps; the gains are not consistent and when they occur they are small. This questions the necessity of an LRM to think before doing MT, all the more so that thinking is considerably more expensive than straight up answering. It is worth noting that "small" models (Qwen-0.6B, Qwen-1.7B and Qwen-4B) often generate answers in English or Chinese when they struggle with the target language (e.g., Czech, Finnish, Kazakh, Lithuanian etc.) resulting in artificially better neural scores. The thinking mode particularly helps in such scenarios because it allows the model to remember that it should write an answer in a different language than what it is "used" to generating in. When the models are big enough (typically ≥ 8B), thereby solving this incorrect language issue, thinking does not result in any gains. Moreover, we run additional experiments with more DeepSeek-R1-Distill models (see Table 4) and again we observe a similar pattern, "no thinking" consistently outperforms thinking. We run additional experiments when translating in the 5-shot setting, retrieving demonstrations from the FLORES 200 dev test with bm25s (Lù, 2024) following Zebaze et al. (2025c). As shown in Table 5, providing demonstrations does not help the thinking mode take performance to an upper level. Similarly, it does not help when translating into English, as reported in Table 6.

## B.2 HUMAN EVALUATION

We carried out a focused human comparison of "thinking" versus "not thinking" for English→French translation using `Qwen3-32B`. We randomly sampled 100 FLORES-200 exam-

---

[7]nrefs:1|case:mixed|eff:yes|nc:6|nw:2|space:no|version:2.4.2

| Models | Czech | | Finnish | | French | | German | | Japanese | |
|---|---|---|---|---|---|---|---|---|---|---|
| | BLEU | MetricX | BLEU | MetricX | BLEU | MetricX | BLEU | MetricX | BLEU | MetricX |
| QWEN3-0.6B | **23.33** | **7.29** | **11.40** | 13.01 | **35.95** | 3.32 | **33.10** | **4.12** | **15.47** | **5.96** |
| *w/o Thinking* | 22.05 | 7.42 | 9.30 | **11.40** | 35.02 | **3.31** | 31.26 | 4.24 | 14.88 | 6.38 |
| QWEN3-1.7B | **34.90** | **3.33** | **23.42** | **6.25** | 41.40 | **1.99** | 40.50 | **2.29** | **23.67** | **3.03** |
| *w/o Thinking* | 34.23 | 3.54 | 22.62 | **6.25** | **41.89** | 2.03 | **40.94** | **2.31** | 22.66 | 3.41 |
| QWEN3-4B | 38.76 | **2.15** | 30.57 | 3.68 | 44.54 | **1.59** | 44.43 | **1.59** | 27.05 | **2.26** |
| *w/o Thinking* | **39.06** | 2.26 | **30.70** | 3.47 | **45.59** | 1.61 | **44.83** | 1.66 | **27.32** | 2.40 |
| QWEN3-8B | 40.02 | **1.80** | 33.28 | 2.52 | 45.80 | **1.42** | 45.14 | 1.50 | 27.98 | **1.92** |
| *w/o Thinking* | **40.77** | 1.86 | **33.49** | **2.46** | **46.54** | 1.43 | **45.46** | **1.42** | **28.95** | 1.93 |
| QWEN3-14B | 41.91 | 1.60 | 35.45 | 2.12 | 46.70 | 1.37 | 46.28 | 1.36 | 29.20 | **1.75** |
| *w/o Thinking* | **43.28** | **1.57** | **36.31** | **2.00** | **48.64** | **1.32** | **47.49** | **1.32** | **30.43** | 1.77 |
| QWEN3-32B | 43.19 | 1.47 | 37.01 | 1.84 | 47.51 | **1.27** | 46.77 | 1.29 | 29.99 | 1.68 |
| *w/o Thinking* | **44.23** | **1.42** | **37.84** | **1.77** | **48.72** | **1.26** | **47.52** | **1.26** | **30.88** | **1.70** |

| Models | Kazakh | | Lithuanian | | Portuguese | | Spanish | | Turkish | |
|---|---|---|---|---|---|---|---|---|---|---|
| | BLEU | MetricX | BLEU | MetricX | BLEU | MetricX | BLEU | MetricX | BLEU | MetricX |
| QWEN3-0.6B | **6.94** | 15.07 | **10.21** | 13.71 | **39.22** | **3.59** | **25.92** | **3.92** | **16.12** | **9.21** |
| *w/o Thinking* | 5.92 | **15.67** | 4.36 | **7.41** | 37.72 | 3.70 | 25.64 | 3.99 | 13.83 | 9.34 |
| QWEN3-1.7B | **17.90** | **8.48** | **23.61** | **6.26** | 45.63 | **2.15** | 29.79 | **2.43** | **29.31** | **4.39** |
| *w/o Thinking* | 16.20 | 9.17 | 22.23 | 6.54 | **46.32** | 2.31 | **30.49** | 2.46 | 27.65 | 4.53 |
| QWEN3-4B | **25.64** | **5.02** | **29.64** | **3.75** | 49.12 | **1.79** | 32.59 | **1.95** | **34.96** | **2.77** |
| *w/o Thinking* | 24.20 | 5.31 | 29.37 | 3.83 | **50.16** | 1.84 | **33.55** | 2.02 | 34.69 | 2.88 |
| QWEN3-8B | **29.31** | **3.80** | 33.01 | **2.76** | 50.04 | **1.54** | 32.89 | **1.75** | **37.88** | **2.15** |
| *w/o Thinking* | 28.91 | 3.98 | **33.28** | 2.81 | **50.89** | 1.56 | **33.80** | 1.78 | 37.77 | 2.24 |
| QWEN3-14B | 32.01 | **3.06** | 34.31 | 2.48 | 51.22 | **1.48** | 34.06 | **1.63** | 39.52 | 1.88 |
| *w/o Thinking* | **32.17** | 3.08 | **35.15** | **2.43** | **53.28** | 1.49 | **35.50** | **1.63** | **40.18** | **1.86** |
| QWEN3-32B | 33.33 | 3.27 | 36.10 | **2.06** | 51.69 | 1.44 | 34.49 | 1.57 | 40.91 | 1.81 |
| *w/o Thinking* | **34.03** | **2.77** | **37.15** | 2.14 | **53.66** | **1.39** | **35.85** | **1.54** | **41.87** | **1.72** |

Table 6: BLEU and MetricX scores for 10 X → English directions from FLORES 200. Best results are highlighted in bold.

ples and carried out reference-free pairwise ranking (better, worse, same) with native French speakers. To avoid biasing the results against the thinking model, we excluded instances where it failed to produce a translation due to getting lost in its CoT. Annotators judged both translations as equally good or bad in 36% of cases; the "thinking" answers were preferred in 31% of cases, and the "not thinking" answers in 33%. We conducted the same experiment in a reference-based fashion to evaluate Turkish→English and found that both translations are equally good or bad in 47% of the cases; "thinking" answers were preferred in 30% of cases, and the "not thinking" answers in 23%. Overall, these findings corroborate our conclusions in Section 3 that MT capabilities of TM do not significantly benefit from thinking tokens.

### B.3 RESULTS ON NTREX-128 AND TICO-19

In this section, we evaluate the models on 2 additional benchmarks:

- **NTREX 128** (Barrault et al., 2019; Federmann et al., 2022) is an MT benchmark derived from WMT19 news data translated by professional human translators. It contains 1997 parallel sentences and is recommended for the evaluation of from-English translation directions. We use the first 1000 sentence pairs for evaluation, and the last 997 sentence pairs as the selection pool.
- **TICO-19** (Anastasopoulos et al., 2020) is an MT benchmark comprising texts on the COVID-19 pandemic covering 35 languages. Its validation and test sets consist of 971 (used as a selection pool) and 2100 samples respectively.

We focus on translating from English. We report the results obtained on NTREX 128 in Table 7 and those obtained on TICO-19 in Table 8. We reach the same conclusions on NTREX 128 than with FLORES 200. On TICO-19 we consider different languages, mostly from Asia. Despite some of them being low-resource languages (Khmer, Marathi, Nepali), thinking offers little to no advantage.

| Models | Czech | | Finnish | | French | | German | | Japanese | |
|---|---|---|---|---|---|---|---|---|---|---|
| | BLEU | MetricX | BLEU | MetricX | BLEU | MetricX | BLEU | MetricX | BLEU | MetricX |
| QWEN3-0.6B | 4.62 | 22.82 | 3.03 | 23.26 | **16.80** | 10.38 | **13.93** | 10.38 | **5.66** | 10.06 |
| *w/o Thinking* | **4.92** | **15.63** | **3.33** | **21.09** | 16.72 | **9.88** | 13.89 | **9.18** | 4.18 | **8.78** |
| QWEN3-1.7B | 13.64 | 16.44 | **6.93** | 19.37 | **26.14** | 5.86 | **24.33** | 4.97 | **12.93** | **6.94** |
| *w/o Thinking* | **14.11** | **16.13** | **6.95** | **19.19** | 25.61 | 6.13 | 23.43 | **4.98** | 12.41 | 7.03 |
| QWEN3-4B | **22.23** | **9.92** | **12.70** | **14.04** | **31.07** | **4.09** | **30.41** | **2.97** | **18.11** | **5.64** |
| *w/o Thinking* | 22.06 | 10.16 | 11.83 | 14.56 | 30.82 | 4.43 | 30.21 | 3.16 | 16.77 | 5.91 |
| QWEN3-8B | **28.40** | **7.19** | **17.02** | **10.28** | 33.10 | **3.51** | 34.19 | **1.98** | 19.55 | **5.09** |
| *w/o Thinking* | 27.54 | 7.92 | 16.63 | 10.82 | **33.44** | **3.51** | **34.27** | 2.20 | **19.99** | 5.15 |
| QWEN3-14B | **31.04** | **5.68** | **20.22** | **7.84** | 34.61 | **3.07** | **36.85** | **1.68** | **22.02** | **4.65** |
| *w/o Thinking* | 29.92 | 6.18 | 19.86 | 8.25 | **34.80** | 3.17 | 36.82 | 1.69 | 21.88 | 4.61 |
| QWEN3-32B | **32.63** | 5.73 | 21.32 | 7.47 | **34.76** | 2.95 | 37.09 | 1.53 | **21.19** | 4.97 |
| *w/o Thinking* | 30.48 | **5.73** | **21.71** | **7.38** | 34.55 | **2.84** | **37.13** | **1.46** | 20.59 | **4.51** |

| Models | Kazakh | | Lithuanian | | Portuguese | | Spanish | | Turkish | |
|---|---|---|---|---|---|---|---|---|---|---|
| | BLEU | MetricX | BLEU | MetricX | BLEU | MetricX | BLEU | MetricX | BLEU | MetricX |
| QWEN3-0.6B | 0.26 | 23.60 | 0.90 | 24.26 | **19.57** | 10.00 | **22.01** | 8.77 | 6.76 | 21.87 |
| *w/o Thinking* | 0.66 | 22.05 | 1.46 | 23.90 | 18.37 | **9.28** | 20.12 | **8.21** | **7.04** | **20.30** |
| QWEN3-1.7B | 0.52 | 23.71 | 4.62 | 21.42 | **29.23** | **5.49** | **30.94** | **4.83** | 12.58 | 16.33 |
| *w/o Thinking* | 1.13 | 23.68 | 5.06 | 21.47 | 28.92 | 5.60 | 30.81 | 4.96 | **12.73** | **15.25** |
| QWEN3-4B | 5.95 | 17.29 | 10.59 | **15.76** | **34.60** | **3.76** | **36.44** | **3.32** | **19.26** | **11.29** |
| *w/o Thinking* | 5.54 | 17.81 | **10.86** | 15.83 | 33.83 | 4.23 | 35.91 | 3.62 | 19.05 | **11.31** |
| QWEN3-8B | **9.70** | **13.39** | 14.80 | 11.97 | 36.71 | **3.10** | **38.91** | **2.78** | 23.07 | **8.67** |
| *w/o Thinking* | 8.93 | 14.07 | **14.96** | **11.92** | **36.89** | 3.35 | 38.85 | 2.95 | **23.29** | 8.84 |
| QWEN3-14B | **13.18** | **10.57** | 18.89 | 9.58 | **38.49** | **2.83** | **40.24** | **2.53** | **26.35** | **7.38** |
| *w/o Thinking* | 12.65 | 11.04 | **19.42** | **9.16** | 38.39 | 2.91 | 40.18 | 2.59 | 26.31 | 7.36 |
| QWEN3-32B | **13.53** | 10.93 | 19.93 | 9.01 | **39.21** | **2.74** | 40.70 | 2.41 | **27.62** | **6.93** |
| *w/o Thinking* | 13.05 | **10.40** | **20.77** | **8.62** | 39.23 | 2.76 | **41.30** | **2.35** | 26.68 | 6.97 |

Table 7: BLEU and MetricX scores for 10 English → X directions from NTREX 128. Best results are highlighted in bold.

| Models | Bengali | | Farsi | | Hindi | | Indonesian | | Khmer | |
|---|---|---|---|---|---|---|---|---|---|---|
| | BLEU | MetricX | BLEU | MetricX | BLEU | MetricX | BLEU | MetricX | BLEU | MetricX |
| QWEN3-0.6B | 0.07 | 22.31 | 1.86 | 20.03 | 0.32 | 21.64 | **22.66** | 7.43 | 0.63 | 22.87 |
| *w/o Thinking* | 0.32 | 21.61 | 3.17 | 3.46 | 0.58 | 21.79 | 21.18 | **6.51** | 1.02 | 23.80 |
| QWEN3-1.7B | 2.85 | 15.49 | 10.07 | **13.50** | 6.32 | 14.70 | 37.04 | **3.47** | 1.04 | 22.10 |
| *w/o Thinking* | **4.69** | **13.13** | **10.46** | 13.56 | **9.03** | 13.15 | **38.33** | 3.45 | 1.54 | 22.21 |
| QWEN3-4B | **14.50** | 6.31 | **19.11** | **7.62** | **24.40** | 6.23 | 45.14 | 2.82 | **9.20** | **15.05** |
| *w/o Thinking* | 13.90 | **6.16** | 18.59 | 7.72 | 23.40 | 6.34 | **46.60** | **2.52** | 8.27 | 15.54 |
| QWEN3-8B | **18.97** | 4.13 | **25.13** | 4.70 | **30.22** | 4.82 | 48.83 | **1.89** | **16.66** | 10.65 |
| *w/o Thinking* | 17.98 | **4.15** | 24.91 | **4.58** | 29.95 | 4.90 | **50.19** | 1.94 | 16.20 | **10.63** |
| QWEN3-14B | **23.68** | 3.10 | 29.01 | 3.55 | **35.81** | **4.12** | 51.21 | 1.66 | 22.09 | **8.21** |
| *w/o Thinking* | 22.87 | 3.28 | **29.33** | 3.38 | 35.65 | 4.26 | **52.07** | 1.71 | **22.30** | 8.41 |
| QWEN3-32B | 17.94 | 7.76 | 27.39 | 4.41 | 31.15 | 6.83 | 51.95 | 1.62 | 16.52 | 11.42 |
| *w/o Thinking* | **24.40** | **2.89** | **29.52** | **3.16** | **37.37** | **3.98** | **52.47** | **1.59** | **21.22** | **7.13** |

| Models | Marathi | | Malay | | Nepali | | Tagalog | | Urdu | |
|---|---|---|---|---|---|---|---|---|---|---|
| | BLEU | MetricX | BLEU | MetricX | BLEU | MetricX | BLEU | MetricX | BLEU | MetricX |
| QWEN3-0.6B | 0.03 | 23.58 | 13.57 | 9.11 | 0.24 | 22.75 | 5.66 | 21.11 | 0.41 | 22.99 |
| *w/o Thinking* | 0.06 | 24.04 | 10.49 | 4.98 | 0.61 | 21.58 | **7.72** | **17.76** | 0.85 | 22.18 |
| QWEN3-1.7B | 1.27 | 19.50 | 22.81 | 4.71 | 2.45 | 17.30 | 10.18 | **18.67** | 2.74 | 18.42 |
| *w/o Thinking* | **2.49** | **18.36** | **23.17** | **4.54** | **9.83** | **10.04** | **10.65** | 18.64 | 3.08 | 18.26 |
| QWEN3-4B | **7.78** | **9.80** | 31.93 | **3.99** | 9.83 | 10.04 | 21.62 | 11.54 | **11.37** | **10.82** |
| *w/o Thinking* | 7.75 | 10.61 | **32.87** | 4.07 | 8.13 | 11.42 | **24.30** | 10.52 | 10.55 | 10.97 |
| QWEN3-8B | **12.01** | **6.92** | 37.30 | **2.95** | **15.25** | **6.93** | 27.44 | 7.60 | 16.35 | 7.06 |
| *w/o Thinking* | 11.93 | 7.03 | **39.25** | 3.12 | 14.42 | 7.53 | **30.00** | 7.18 | **16.56** | **6.88** |
| QWEN3-14B | 15.05 | **5.35** | 41.91 | **2.63** | **18.23** | **5.89** | 32.38 | **5.35** | 20.97 | **5.06** |
| *w/o Thinking* | **15.11** | 5.49 | **42.91** | 2.67 | 17.93 | 6.26 | **34.29** | 5.33 | 21.04 | 5.24 |
| QWEN3-32B | 13.26 | 8.61 | 43.30 | **2.57** | 17.40 | 8.31 | 31.44 | 6.04 | 20.21 | 6.36 |
| *w/o Thinking* | **16.63** | **4.98** | **44.12** | 2.65 | **20.83** | **5.45** | **36.12** | **4.80** | **21.87** | **4.87** |

Table 8: BLEU and MetricX scores for 10 English → X directions from TICO-19. Best results are highlighted in bold.

### B.4  DOES DISTILLED CHAIN-OF-THOUGHT AS INTERMEDIATE TOKENS IMPROVE PERFORMANCE?

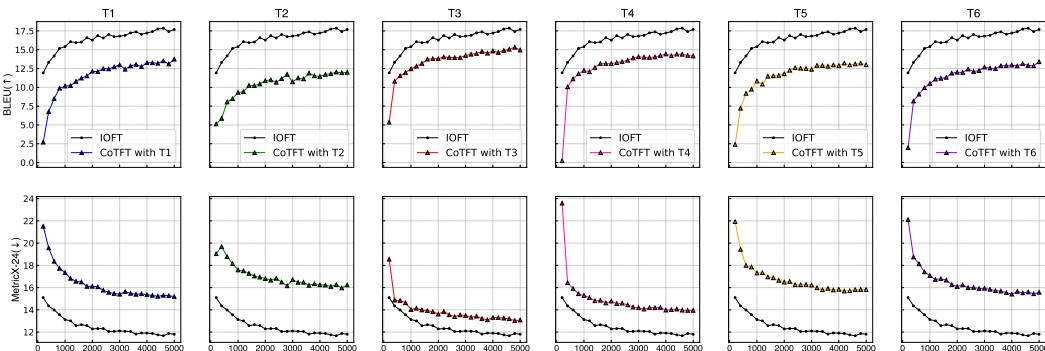

Figure 8: Comparison between COTFT and IOFT with six different CoT templates.

Following Section 5.2, we compare COTFT with IOFT across all six templates using `gemma-3-1b-pt` as the student and `gemma-3-27b-it` as the intermediate teacher. We focus on translating from English to Lithuanian. As shown in Figure 8, COTFT consistently lags behind IOFT. The gap can be as large as 5 BLEU and 4 MetricX. Despite T3 being the best template, it is still largely behind IOFT in terms of performance.

### B.5  WHAT HAPPENS WHEN WE USE TRACES FROM MT PROMPTING STRATEGIES AS INTERMEDIATE TOKENS?

In Figure 9, we observe that COTFT with reasoning traces based on alternative prompting strategies outperforms IOFT. SBYS is an exception, for which COTFT is behind IOFT. Across prompting strategies, IOFT-MAX outperforms IOFT and COTFT with the only exception of CompTra. This is exactly what happened with our experiments with `Llama-4-Scout-17B-16E-Instruct` and `gemma-3-4b-pt` in Xhosa.

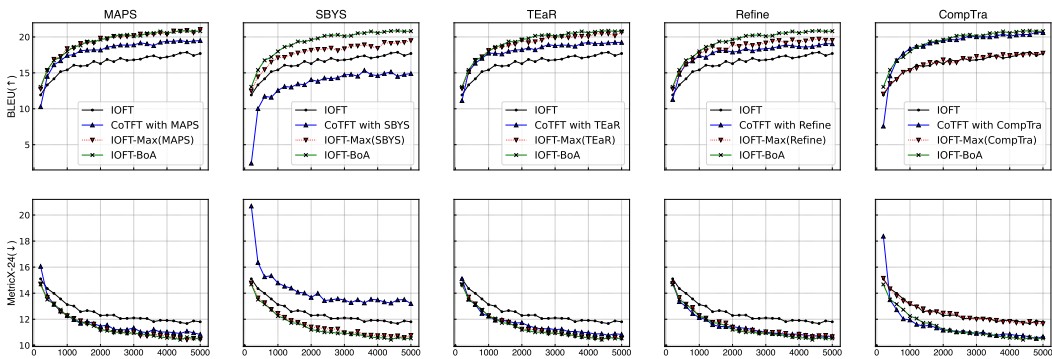

Figure 9: Comparison between IOFT and COTFT with five different prompting strategies.

Finally, as shown in Figure 10, COTFT-MAX fails to improve over IOFT-MAX, confirming our previous conclusions with Xhosa. In this case, CompTra is not an exception. For all the strategies, COTFT-MAX and IOFT-MAX are very close in performance, with IOFT-BOA topping them all. This again suggests that reasoning traces do not help, even when they are based on MT prompting strategies whose drafting attempts do not outperform the ground truth in terms of quality. Having target translations of high-quality (IOFT-BOA) has the highest impact, outperforming the standard IOFT by 3 BLEU and 1.3 MetricX with the same number of parallel pairs and the same training recipe.

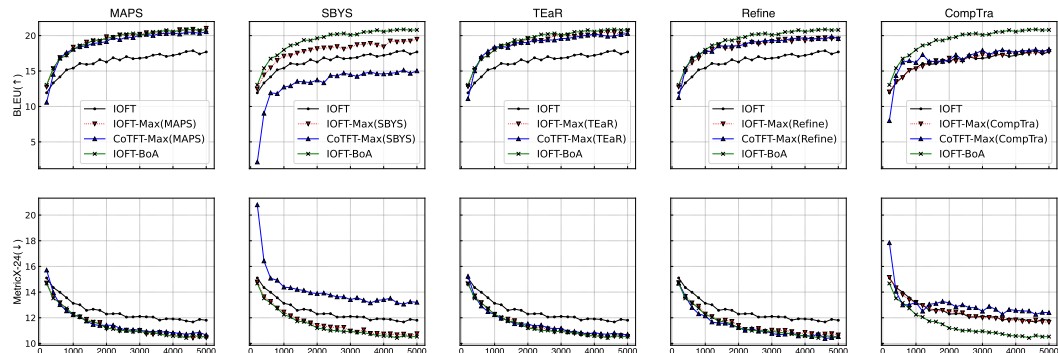

Figure 10: Comparison between IOFT and CoTFT with five different prompting strategies.

## B.6 Down the rabbit hole of sentence decomposition

Following Section 6.1, we evaluate multiple sentence decomposition approaches and compare CoTFT against standard IOFT and IOFT-Ext. As shown in Figure 11, CoTFT consistently outperforms IOFT across all decomposition strategies. Again, SP and CompTra works better than P and H. Using the generated pairs as additional training samples (i.e. IOFT-Ext) is particularly helpful with P and SP because they correspond to fully-fledged sentences as explained earlier. CoTFT with H and CoTFT with CompTra outperform the corresponding IOFT-Ext suggesting that short phrases and their translations are relevant intermediate information for CoTFT. CoTFT with CompTra works just as well as IOFT-Ext(P) which is impressive as the latter required multiplying the size of $\mathcal{D}$ by six.

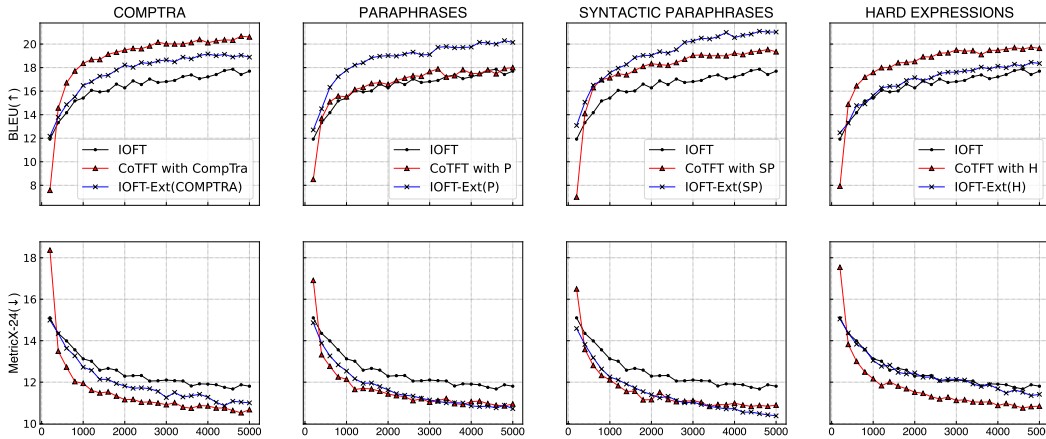

Figure 11: Comparison between IOFT and CoTFT with four different sentence decomposition strategies.

## B.7 CoT distillation: What happens when we change the teacher?

In this section, we run the same CoT distillation experiment as in Section 5.2 but we use `DeepSeek-R1-Distill-Llama-70B` instead as the teacher. As seen in Figure 12, CoTFT behaves similarly to IOFT across templates. The performance of CoTFT is better than what we observed with `Llama-4-Scout-17B-16E-Instruct` despite LLaMA being better at translating into Xhosa. We attribute this to the "thinking" abilities of DeepSeek-R1 which, despite not being good at generating Xhosa, can generate a better explanation compared to LLaMA as to why a hypothesis is an accurate translation of a source. However, in both cases, CoTFT does not improve over IOFT, which is also faster to train in comparison.

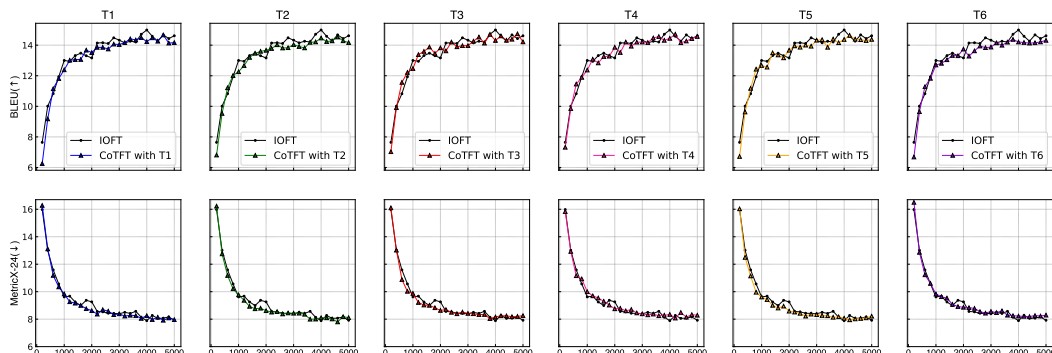

Figure 12: Comparison between IOFT and COTFT with six different CoT templates.

### B.8 MT TRACES GENERATED BY PROMPTING STRATEGIES AS INTERMEDIATE TOKENS: WHAT HAPPENS WHEN WE CHANGE THE TEACHER?

We run additional information on Xhosa and change the teacher from `Llama-4-Scout-17B-16E-Instruct` to `gemma-3-27b-it`. GEMMA's zero-shot MT performance (on FLORES 200, BLEU = 12.82, MetricX = 7.62) is worse than LLAMA's (BLEU = 16.90, MetricX = 6.53) and we aim to investigate how this impacts our findings. First of all, COTFT outperforms IOFT across prompting strategies with the exception of SBYS. IOFT-MAX outperforms IOFT and we observe the same behaviour between IOFT-MAX(COMPTRA) and IOFT as we did with LLAMA. Despite COTFT with SBYS underperforming IOFT, IOFT-MAX(SBYS) outperforms IOFT, meaning that translation attempts embedded in SBYS-inspired CoT are helpful, but they are drowned out by other useless tokens, which impacts how well COTFT performs. Ultimately, IOFT-BOA works best; although IOFT-MAX(TEAR) achieves higher BLEU scores it lags behind in terms of MetricX. It even outperforms the teacher `gemma-3-27b-it` despite being six times smaller.

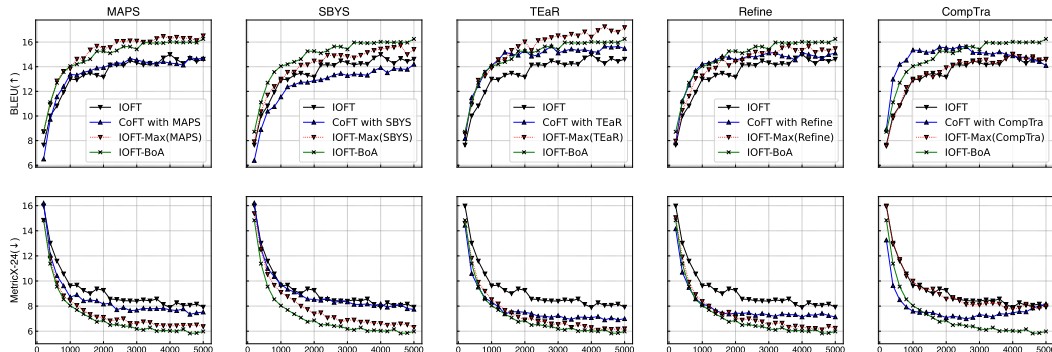

Figure 13: Comparison between IOFT and COTFT with five different prompting strategies.

When we use `gemma-3-4b-it` as the teacher (`gemma-3-4b-pt` being the student), the traces obtained using prompting strategies do not help COTFT to outperform IOFT. In Figure 14, we observe a degradation of performance that confirms our intuition suggesting that these traces are helpful only if they contain translation attempts that are better than the ground truth.

### B.9 REINFORCEMENT LEARNING AFTER IOFT AND COFT

Building on the experiments presented in Section 6.2, we apply GRPO to the final checkpoints (checkpoint-5000) under three additional configurations: COTFT with MAPS, SBYS, TEaR, and Self-Refine. The results, shown in Figure 15, indicate that GRPO results in consistent improvements of approximately +1 BLEU and -0.7 MetricX points across all setups, mirroring the trends observed

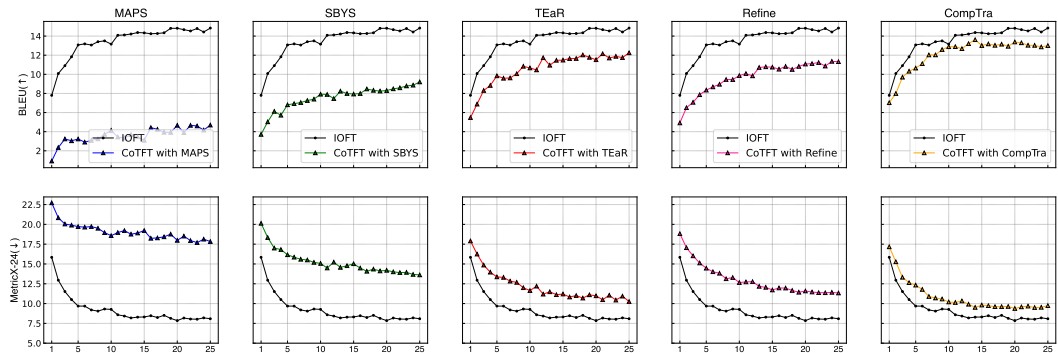

Figure 14: Comparison between IOFT and CoTFT with five different prompting strategies.

with CompTra. Notably, GRPO maintains the relative performance ordering between IOFT and CoTFT prior to fine-tuning. However, CoTFT models do not gain more from GRPO than IOFT models, and in practice, performing IOFT alone (rather than GRPO) can achieve comparable or greater gains at a lower computational cost.

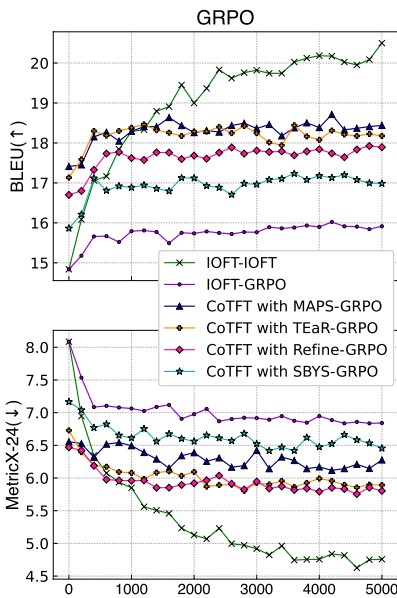

Figure 15: Comparison between IOFT and CoTFT with GRPO.

### B.10 WHAT HAPPENS WHEN WE CHANGE THE STUDENT?

We conduct additional English→Xhosa experiments with `Llama-4-Scout-17B-16E-Inst` as the teacher, but replace the student model `gemma-3-4b-pt` with `gemma-3-12b-pt`. To reduce computational costs, we apply LoRA (Hu et al., 2022) while keeping all other hyperparameters unchanged. Overall, CoTFT continues to underperform IOFT across all six CoT distillation templates (Figure 16). The performance gap is noticeably larger than with `gemma-3-4b-pt`, which we attribute to the use of LoRA. Conversely (Figure 17), traces derived from MT prompting strategies generally serve as better intermediate tokens for CoTFT: CompTra, Self-Refine, and MAPS all surpass IOFT. As before, IOFT-MAX outperforms standard IOFT, and the same pattern holds when comparing IOFT-MAX(COMPTRA) to IOFT, mirroring our observations with the 4B model. Even though CoTFT with SBYS and TEaR underperforms IOFT, their IOFT-MAX variants (IOFT-MAX(SBYS) and IOFT-MAX(TEAR)) achieve higher performance. This suggests that the trans-

lation attempts embedded within SBYS and TEaR-style CoT are indeed useful, but their signal is diluted by extraneous tokens when used directly in COTFT. In line with the rest of our findings, IOFT-BOA achieves the strongest results.

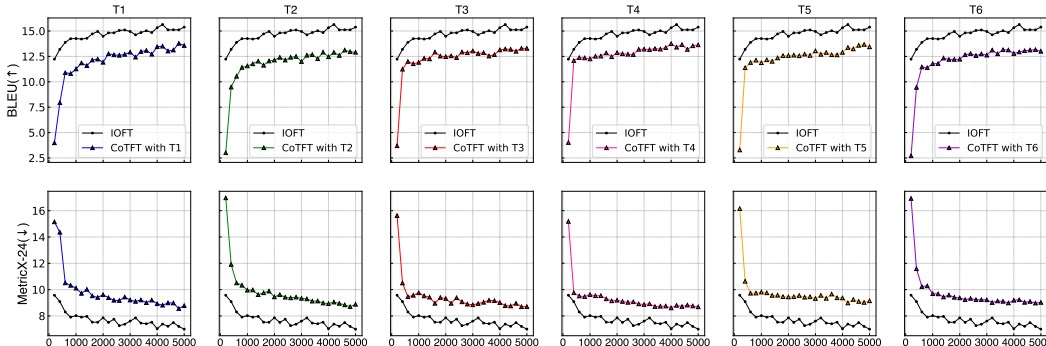

Figure 16: Comparison between IOFT and COTFT with six different CoT templates (gemma-3-12b-pt).

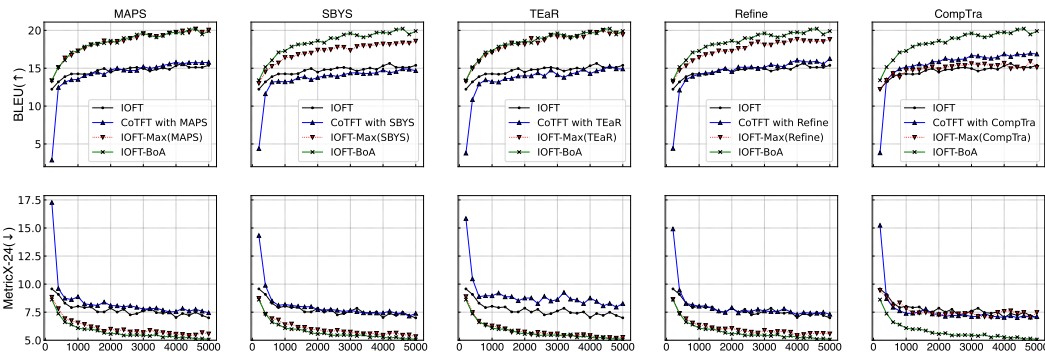

Figure 17: Comparison between IOFT and COTFT with five different prompting strategies (gemma-3-12b-pt).

## C ABLATION STUDIES AND EXAMPLES OF PROMPTS

### C.1 ABLATION STUDIES

In this section, we analyze how our hyperparameter choices influence the observed results. We conduct three main experiments:

- **Increasing the number of IOFT steps.** Rather than training for 5000 steps as described in Section 5.1, we extend IOFT training to 10000 steps to assess whether additional optimization results in further improvements.
- **Increasing gradient accumulation for COTFT.** This experiment shares the same objective as the first. Beyond simply extending the number of training steps, increasing the effective batch size is another way to scale training. Given our computational budget, we opt for the latter and study how varying the number of gradient accumulation steps affects CoT distillation.
- **Scaling GRPO.** We adjust the configuration from Appendix A.5 by running GRPO for 10000 steps and generating 24 samples per prompt.

In addition to these experiments, we evaluate GRPO with an auxiliary reward. Starting from a model SFT-trained via COTFT with CompTra, we introduce a reward based on the average BLASER-2.0

QE score computed between the phrase–translation pairs produced during generation. All other factors are held constant to isolate the effect of this reward on GRPO performance. Figure 18 summarizes the results of these four experiments. For SFT, increasing training scale provides minimal benefits, indicating that our existing configuration is already adequate. This also suggests that the advantage of IOFT is not due to insufficient CoTFT training. In contrast, scaling RL fine-tuning provides modest improvements. Most of the gains occur early in training (before step 5000), but extending training further still leads to slight additional improvements, particularly on MetricX, while BLEU remains largely unchanged.

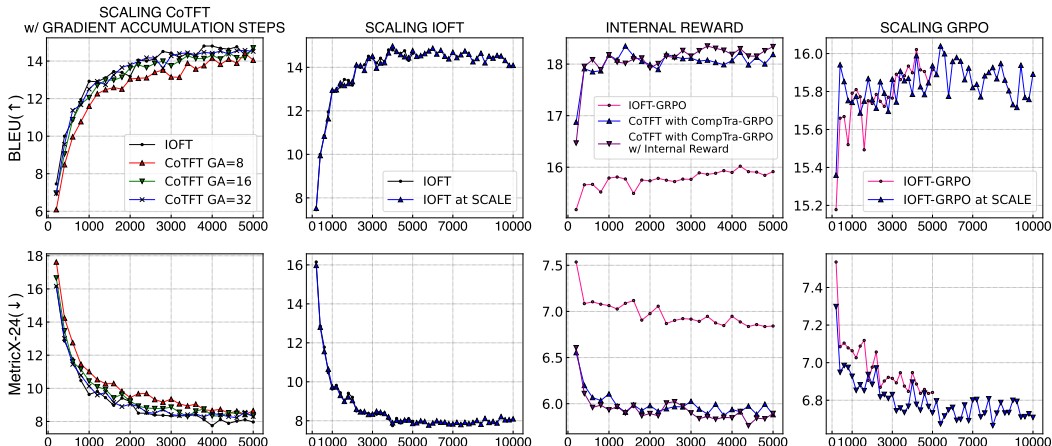

Figure 18: Ablation studies on the hyperparameters. The x-axis represents the number of steps.

We also assess the diversity of generations produced after each fine-tuning strategy. To do so, we compile all strategies into a BLEU-based similarity matrix, where each entry reflects the word-level similarity between the FLORES-200 outputs of a pair of methods (applied to `gemma-3-4b-pt`). As shown in Figure 19, CoT distillation produces more similar generations across templates than any other fine-tuning approach. We observe the same pattern for CoTFT across different prompting strategies. IOFT is most similar to IOFT-MAX(COMPTRA), likely because BLASER-2.0 QE frequently prefers the ground-truth translation over the translation attempts embedded with CompTra's traces (see Section 4). It is also close to IOFT-EXT(COMPTRA) and IOFT-EXT(H), supporting our intuition that short phrases and hard expressions are not ideal as additional training examples (Section 6.1). Overall, BLEU scores rarely exceed 50, indicating substantial reformulation across strategies. While overlaps do exist, even strategies achieving very similar MT performance often produce noticeably different generations.

## C.2 EXAMPLES OF TRAINING SAMPLES

During CoT distillation, we prompt a teacher (e.g. `Llama-4-Scout-17B-16E-Instruct`) with a source-target pair and ask it to produce a reasoning explaining how to derive the target given the source. Here are some examples with source = "Agdistis was a Phrygian deity known for his extraordinary powers." and target (in Xhosa) = "Agdistis yayingumthandazeli wamaPhrygia owaziwayo ngamandla akhe angaqhelekanga." for each of the CoT templates T1-T6.

$T_1$

```
<think>
I am analyzing the sentence structure of the source sentence and
    identifying the core elements, which are "Agdistis" (subject), "was"
    (verb), and "a Phrygian deity known for his extraordinary powers" (
    object).
I am recognizing that "Agdistis" is a proper noun and will likely remain
    unchanged in the target sentence.
I am identifying the key components of the object: "Phrygian deity" and "
    extraordinary powers".
```

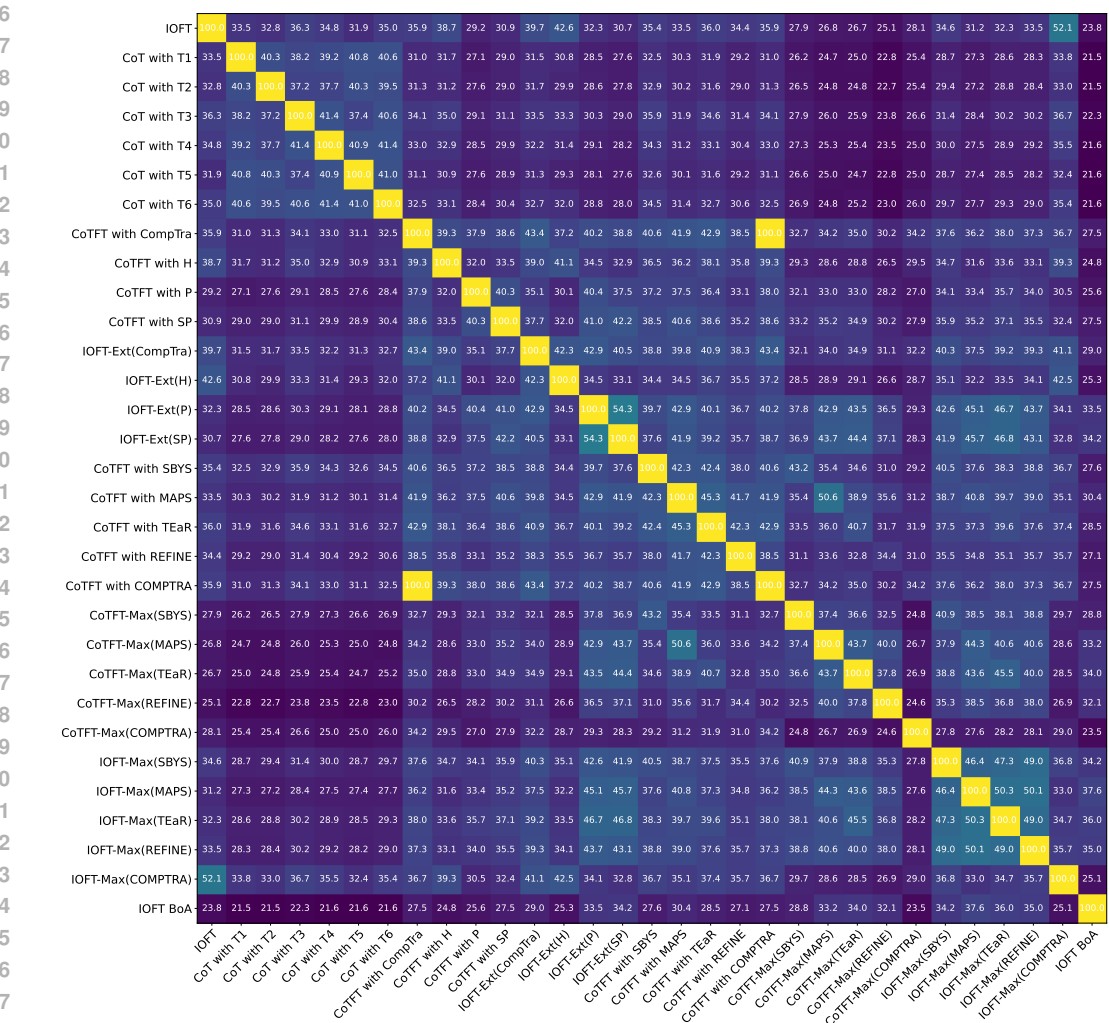

Figure 19: BLEU scores between the FLORES-200 generations of each pair of methods.

```
I am translating "Agdistis" to Xhosa, which remains "Agdistis" since it's
    a proper noun.
I am translating "was" to Xhosa, which is "yaying" (a form of the verb "
    to be" in Xhosa).
I am translating "a Phrygian deity" to Xhosa, which becomes "umthandazeli
    wamaPhrygia" (umthandazeli meaning deity or god, and wamaPhrygia
    meaning of Phrygia).
I am translating "known for his extraordinary powers" to Xhosa, which
    becomes "owaziwayo ngamandla akhe angaqhelekanga" (owaziwayo meaning
    known for, ngamandla akhe meaning his powers, and angaqhelekanga
    meaning extraordinary or unusual).
I am combining the translated elements to form a basic translation: "
    Agdistis yayingumthandazeli wamaPhrygia owaziwayo ngamandla akhe
    angaqhelekanga".
I am reviewing the translation for basic accuracy and grammatical
    structure, and it seems to align with the target sentence.
I am checking for areas that need further refinement, such as word choice
    , tense, or word order, but the translation appears to be accurate.
I am finalizing the translation, ensuring it retains the original meaning
    while improving readability, and the translation matches the target
    sentence.
```

```
</think>
```

$T_2$

```
<think>
I am identifying the basic elements of the source sentence: The subject
    is "Agdistis", the verb is "was", and the object is "a Phrygian deity
     known for his extraordinary powers".

I am breaking down the object into its components: "a Phrygian deity" is
    a noun phrase describing the type of deity, and "known for his
    extraordinary powers" is a dependent clause describing the deity.

I am converting these elements into an intermediate language structure: I
     have "Agdistis" as the subject, "was" as the verb, and "a Phrygian
    deity" and "known for his extraordinary powers" as descriptors.

I am translating "Agdistis" to Xhosa, which remains "Agdistis" since it's
     a proper noun.

I am translating "was" to Xhosa, which can be "yayingumthandazeli" or a
    similar linking verb, but I will refine this later.

I am translating "a Phrygian deity" to Xhosa, which becomes "umthandazeli
     wamaPhrygia".

I am translating "known for his extraordinary powers" to Xhosa, which
    becomes "owaziwayo ngamandla akhe angaqhelekanga".

I am combining these elements into a rough Xhosa sentence: "Agdistis
    yayingumthandazeli wamaPhrygia owaziwayo ngamandla akhe
    angaqhelekanga".

I am checking for accuracy: The noun-verb agreement seems correct, and
    the connectors appear to be in order.

I am adjusting the word order to ensure it aligns with Xhosa grammatical
    structure: The sentence seems to follow a subject-verb-object word
    order, which is correct for Xhosa.

I am reviewing the translation for naturalness, idiomatic use, and
    overall flow: The sentence seems natural and idiomatic in Xhosa.

I am confirming that the translation "Agdistis yayingumthandazeli
    wamaPhrygia owaziwayo ngamandla akhe angaqhelekanga" accurately
    conveys the meaning of the source sentence.
</think>
```

$T_3$

```
<think>
I am analyzing the provided context in the source language, which is the
    English sentence: "Agdistis was a Phrygian deity known for his
    extraordinary powers."

I identify the key elements in the source sentence: "Agdistis" (proper
    noun), "Phrygian deity" (noun phrase), and "known for his
    extraordinary powers" (adjectival phrase describing the deity).

I am now translating the source text to the target language, which is
    Xhosa.

I start by translating "Agdistis" to Xhosa, which seems to remain "
    Agdistis" as it is a proper noun.
```

```
I then translate "was a Phrygian deity" to Xhosa. "Deity" translates to "
    umthandazeli" and "Phrygian" to "wamaPhrygia", so this part becomes "
    yayingumthandazeli wamaPhrygia".

Next, I translate "known for his extraordinary powers". "Known for"
    translates to "owaziwayo ngamandla", and "extraordinary powers" to "
    angaqhelekanga", so this part becomes "owaziwayo ngamandla akhe
    angaqhelekanga".

I combine these translations to get the target sentence: "Agdistis
    yayingumthandazeli wamaPhrygia owaziwayo ngamandla akhe
    angaqhelekanga".

I perform back translation from the target language to the source
    language to verify accuracy.

The target sentence "Agdistis yayingumthandazeli wamaPhrygia owaziwayo
    ngamandla akhe angaqhelekanga" back-translates to: "Agdistis was a
    Phrygian deity known for his extraordinary powers".

I compare the back translation with the original source context and see
    that they match.

I evaluate whether the meaning of the back translation aligns with the
    original and find that it does.

Since there are no discrepancies identified, I finalize the translation.

I ensure both forward and back translations accurately align across all
    languages involved.

The translation process confirms that "Agdistis yayingumthandazeli
    wamaPhrygia owaziwayo ngamandla akhe angaqhelekanga" is an accurate
    translation of "Agdistis was a Phrygian deity known for his
    extraordinary powers".
</think>
```

$T_4$

```
<think>
I am analyzing the current sentence, "Agdistis was a Phrygian deity known
     for his extraordinary powers," along with the fact that there are no
     previous sentences, to understand that this is a standalone
    statement about a historical or mythological figure.
I identify that the tone is formal and the subject matter is mythological
     or historical, indicating a need for a formal and respectful tone in
     the translation.
I am translating the sentence, starting with the name "Agdistis," which
    seems to be a proper noun and will likely remain unchanged or
    transliterated in the target language.
I recognize the phrase "was a Phrygian deity" as a descriptor of Agdistis
    's origin or classification, which will need to be translated into
    Xhosa in a way that respects the formal tone and accurately conveys
    the meaning.
I note the phrase "known for his extraordinary powers," which indicates a
     need to translate "extraordinary powers" in a way that is
    understandable and natural in Xhosa.
I am considering how to translate "Agdistis was a Phrygian deity" into
    Xhosa, thinking about how to express "deity" and "Phrygian" in the
    target language, likely using "umthandazeli" for deity and "
    wamaPhrygia" for Phrygian.
I am translating "known for his extraordinary powers" by thinking about
    how to say "extraordinary powers" in Xhosa, considering "amandla akhe
     angaqhelekanga" as a possible translation.
```

```
I refine my translation to ensure it fits the context, deciding on "
    Agdistis yayingumthandazeli wamaPhrygia owaziwayo ngamandla akhe
    angaqhelekanga" as a coherent and natural-sounding sentence in Xhosa.
I verify that "Agdistis yayingumthandazeli wamaPhrygia owaziwayo
    ngamandla akhe angaqhelekanga" maintains coherence as a standalone
    sentence and ensures consistency in language and tone, which appears
    to be formal and informative.
I finalize the translation by cross-checking it with the context, which
    is essentially nonexistent, and decide that "Agdistis
    yayingumthandazeli wamaPhrygia owaziwayo ngamandla akhe
    angaqhelekanga" feels natural and appropriately aligned for a formal
    discussion about mythological figures.
</think>
```

$T_5$

```
<think>
I am analyzing the source sentence "Agdistis was a Phrygian deity known
    for his extraordinary powers" to identify the key elements, which
    include the subject "Agdistis", the verb "was", and the descriptive
    phrase "a Phrygian deity known for his extraordinary powers".

I am determining that the most suitable translation strategy for this
    sentence is likely to be a mix of literal and idiomatic translation,
    as I need to convey the specific cultural and mythological references
     accurately while also ensuring the sentence flows well in Xhosa.

I am selecting the translation for "Agdistis" as "Agdistis" since proper
    nouns are often not translated, and it will be retained in its
    original form in Xhosa.

I am choosing to translate "was" into Xhosa as "yayingum", which is a
    linking verb used to describe a state of being, fitting for
    describing someone's identity or nature.

I am translating "a Phrygian deity" into Xhosa as "umthandazeli
    wamaPhrygia", where "umthandazeli" means deity or god, and "
    wamaPhrygia" specifies the region or origin, adapting to Xhosa's
    grammatical structure.

I am deciding to translate "known for his extraordinary powers" as "
    owaziwayo ngamandla akhe angaqhelekanga", where "owaziwayo" means
    known, "ngamandla" means powers, "akhe" signifies possession (his),
    and "angaqhelekanga" means extraordinary or unusual, ensuring the
    descriptive phrase conveys the intended meaning.

I am reviewing my translation decisions and making adjustments for
    fluency and accuracy, ensuring that the Xhosa sentence "Agdistis
    yayingumthandazeli wamaPhrygia owaziwayo ngamandla akhe
    angaqhelekanga" accurately reflects the source sentence.

I am explaining that I chose specific words or phrases to ensure that the
     translation is not only accurate but also natural-sounding in Xhosa,
     balancing between literal translation and contextual appropriateness
    .

I am providing a final explanation for my translation choices, noting
    that while "Agdistis" remains unchanged, other parts of the sentence
    were adapted to fit Xhosa grammar and vocabulary, ensuring the target
     sentence effectively communicates the information from the source
    sentence.

I am concluding that my translation decisions aimed to maintain the
    original meaning while making necessary adjustments for linguistic
    and cultural differences between English and Xhosa.
```

```
</think>
```

$T_6$

```
<think>
I am analyzing the sentence's syntactic structure in the source language,
    and I see that "Agdistis was a Phrygian deity known for his
    extraordinary powers" is in the passive voice, with "Agdistis" as the
    subject, "was" as the linking verb, and "a Phrygian deity known for
    his extraordinary powers" as the predicate nominative.

I am determining the most appropriate syntactic structure in the target
    language, and I think I can maintain a similar structure in Xhosa,
    but I need to consider the language's own grammatical rules and word
    order.

I am adjusting the word order and grammatical structure in the target
    language to match the sentence's meaning, while maintaining clarity,
    and I note that Xhosa typically uses an SVO (Subject-Verb-Object)
    word order.

I am translating the sentence, ensuring that subject-verb-object
    relationships and other syntactic elements align with target language
    norms, and I start with "Agdistis" as the subject, which becomes "
    Agdistis" in Xhosa.

I am translating the linking verb "was" and consider using the Xhosa
    equivalent "yayingum" (a form of "to be").

I am translating the predicate nominative "a Phrygian deity" and consider
    using "umthandazeli wamaPhrygia" (a Xhosa phrase for "a Phrygian
    deity").

I am translating the phrase "known for his extraordinary powers" and
    consider using "owaziwayo ngamandla akhe angaqhelekanga" (a Xhosa
    phrase that maintains the same meaning).

I am checking the sentence's grammar and overall flow in the target
    language, and I have "Agdistis yayingumthandazeli wamaPhrygia
    owaziwayo ngamandla akhe angaqhelekanga".

I am refining the structure by adjusting word choice or reordering
    components to ensure the sentence feels natural and fluid in Xhosa,
    and I think "Agdistis yayingumthandazeli wamaPhrygia owaziwayo
    ngamandla akhe angaqhelekanga" meets these criteria.

I am confirming that the translation accurately conveys the original
    meaning and is clear and fluid in the target language, and I am
    satisfied with the result.
</think>
```

For MT prompting strategies, we first prompt the teacher with the source sentence to generate all intermediate steps required by each strategy. We then concatenate these steps into a single training sample while keeping the original ground-truth target unchanged. Below, we provide example training samples for the five strategies used in this work.

**SBYS**

```
<think>
During the research phase, I have identified the following potential
    translation challenge:

* Idiomatic Expressions:
```

```
There are no apparent idiomatic expressions in the given context text: "
    Agdistis was a Phrygian deity known for his extraordinary powers."
    However, the phrase "extraordinary powers" may pose a challenge as it
     is an idiomatic expression that may not have a direct equivalent in
    Xhosa.

In English, "extraordinary powers" is an idiomatic expression used to
    describe exceptional or supernatural abilities. To translate this
    phrase into Xhosa, I may need to use a more literal translation or
    find an equivalent Xhosa expression that conveys the same meaning.

Additionally, I have noted that:

* "Phrygian" refers to an ancient region in Asia Minor, and the term may
    not be well-known in Xhosa culture. This may require some explanatory
     notes or a more detailed translation to convey the cultural and
    historical context.

No other idiomatic expressions or potential translation challenges were
    identified in the given context text.

Please let me know if I should proceed with the next stage of the
    translation process.

UAgdistis waye uThixo wasePhrygia owaziwa ngamandla akhe angaqhelekanga.

Here's a brief explanation of my translation choices:

* "Agdistis" is left untranslated as it is a proper noun and a name of a
    deity.
* "was a" is translated to "waye u" which is a Xhosa copular phrase used
    to describe a person's identity or status.
* "Phrygian" is translated to "wasePhrygia" which indicates the region or
     origin of the deity.
* "deity" is not directly translated, but implied in the context of "
    UThixo" which means "God" or "deity".
* "known for" is translated to "owaziwa ngamandla" which means "known for
     his powers".
* "extraordinary powers" is translated to "angaqhelekanga" which is an
    adverb that means "unusual", "exceptional" or "remarkable", and "
    amandla" which means "powers".

Please let me know if I should proceed with the next stage of the
    translation process.

Now let's move to the next stage: Post-editing with local refinement.
In this stage, the primary aim is to refine the draft translation by
    making micro-level improvements that improve the draft's fluency.

Here is a refined version of the translation
UAgdistis waye uThixo wasePhrygia owaziwa ngamandla akhe ayengavamile.

Now, we will proofread the refined text for grammar spelling, punctuation
    , terminology and overall fluency.

Here is the translation after proofreading
After proofreading the refined translation, I suggest the following final
    , polished translation:

We will further improve it to obtain the final, polished translation.
</think>

Final Translation
Agdistis yayingumthandazeli wamaPhrygia owaziwayo ngamandla akhe
    angaqhelekanga.
```

**MAPS**

```
<think>
Here is a draft translation

1. UAgdistis waye uhlambe wasePhrygia owaziwa ngamandla akhe
    angaqhelekanga.

Let's write an English sentence related to but different from the input
    English sentence and translate it into Xhosa

The cult of Agdistis, a Phrygian deity, was centered there, and the site
    was probably a major religious center. Isikhululo sika Agdistis,
    isithixo sasePhrygian, sasigxile apho, yaye indawo yayo yayisikhungo
    esikhulu senkonzo yezokholo.

Given this knowledge, we can draft another translation

2. Agdistis wayesithixo sasePhrygian esaziwa ngamandla akhe
    angaqhelekanga.

Let's extract the keywords in the provided English sentence, and then
    translate these keywords into Xhosa

* Agdistis=Agdistis
* Phrygian=Frigiya
* deity=ility
* extraordinary=angakhethekileyo
* powers=amandla

Given this knowledge, we can draft another translation

3. UAgdistis wayesithixo saseFrigiya esaziwa ngamandla akhe
    angakhethekileyo.

Let's use a few words to describe the topics of the provided English
    sentence

Mythology, ancient deity

Given this knowledge, we can draft another translation

4. Agdistis wayeyona dea yasePhrygian eyaziwa ngamandla ayo
    angaqhelekanga.

We will choose the best of these translations and further improve it to
    obtain the final, polished translation.
</think>

Final Translation
Agdistis yayingumthandazeli wamaPhrygia owaziwayo ngamandla akhe
    angaqhelekanga.
```

**TEaR**

```
<think>
Here is a draft translation

1. Agdistis wayesithixo sasePhrygia esaziwayo ngamandla akhe
    angaqhelekanga.

Let's identify errors and assess the quality of the draft translation.
```

```
The categories of errors are accuracy (addition, mistranslation, omission
    , untranslated text), fluency (character encoding, grammar,
    inconsistency, punctuation, register, spelling), locale convention (
    currency, date, name, telephone, or time format) style (awkward),
    terminology (inappropriate for context, inconsistent use), non-
    translation, other, or no-error.
Each error is classified as one of three categories: critical, major, and
    minor. Critical errors inhibit comprehension of the text. Major
    errors disrupt the flow, but what the text is trying to say is still
    understandable. Minor errors are technical errors but do not disrupt
    the flow or hinder comprehension.

Here are the MQM annotations of the draft:
critical: no-error
major: no-error
minor: no-error

Upon reviewing the translation and error information, we can refine the
    draft and obtain a better translation

2. Agdistis wayesithixo sasePhrygia esaziwayo ngamandla akhe angavamile.

We will further improve it to obtain the final, polished translation.
</think>

Final Translation
Agdistis yayingumthandazeli wamaPhrygia owaziwayo ngamandla akhe
    angaqhelekanga.
```

**Self-Refine**

```
<think>
Here is a draft translation

1. UAgdistis waye uhlambe wasePhrygia owaziwa ngamandla akhe
    angaqhelekanga.

Let's improve it and write a better translation

2. UAgdistis waye uhlambe wasePhrygia owaziwa ngamandla akhe amakhulu.

Let's further improve it and write a better translation

3. UAgdistis waye uhlambe wasePhrygia owaziwa ngamandla akhe
    angaqhelekanga.

Let's improve it one last time and write a better translation

4. UAgdistis waye uhlambe wasePhrygia owaziwa ngamandla akhe amakhulu.

We will choose the best of these translations and further improve it to
    obtain the final, polished translation.
</think>

Final Translation
Agdistis yayingumthandazeli wamaPhrygia owaziwayo ngamandla akhe
    angaqhelekanga.
```

**CompTra**

```
<Demonstrations>
1. English sentence
Agdistis was a Phrygian deity.
Xhosa translation
```

```
UAgdistis wayesithixo sasePhrygia.

2. English sentence
Agdistis was known for his extraordinary powers.
Xhosa translation
Agdistis waziwa ngamalawulo akhe angaqhelekanga.
</Demonstrations>

Final Translation
Agdistis yayingumthandazeli wamaPhrygia owaziwayo ngamandla akhe
    angaqhelekanga.
```

