# OpenReview forum: "LLM Reasoning for Machine Translation: Synthetic Data Generation over Thinking Tokens"
_ICLR.cc/2026/Conference — Submitted to ICLR 2026_

### Official Review · Reviewer_ErSk · 2025-10-31

**Soundness:** 2
**Presentation:** 3
**Contribution:** 2
**Rating:** 4
**Confidence:** 4

**Summary:**

This paper investigates whether thinking and CoT (w/ and w/o distillation) can improve machine translation. The authors evaluate multiple models (Qwen3, Llama-4, Gemma) and language pairs with various settings to compare “thinking” vs “non-thinking”, and conduct CoT finetuning where a teacher model’s reasoning traces are distilled into a student model. The results show that neither generating thinking tokens nor distilling CoT improves translation quality. Instead, the key driver of improvement is the presence of parallel translation attempts within the data. The paper concludes that enhancing translation data quality and quantity is more effective.

**Strengths:**

- Unlike prior CoT studies that mainly focused on mathematical or logical reasoning tasks, this paper systematically extends and verifies the approach in the domain of machine translation.

- The experiments are extensive, and the figures (Figure 1–7) and tables (e.g., Table 1) are well-structured, making the experimental design and logical flow easy to understand.

- By demonstrating that data quality and quantity matter more than the mere presence of “thinking”, the paper offers a practical and data-centric perspective for future LLM-based translation research.

**Weaknesses:**

- As a research paper, the analysis of why CoT is ineffective in MT remains superficial. Beyond the numerical evidence showing that CoT provides no gain, a deeper discussion of what structural differences between MT and reasoning tasks lead to this outcome, or a qualitative comparison of cases where CoT helps vs. where it fails, would strengthen the argument.

- (Minor) While there are no severe grammatical mistakes, there are occasional stylistic and typographical inconsistencies. For instance, in lines 416–417, “S” should be “P”, and in lines 350–352, the use of appropriate conjunctions would make the sentence flow more naturally.

**Questions:**

- Table 1 presents results across multiple languages, model sizes, and experimental conditions, but the analysis concludes rather briefly that CoT has no effect. Could the authors test this further in more extreme scenarios, for instance, with smaller models ( < 0.6 B ) whose capabilities might collapse without CoT, or in low-resource language-to-language translation settings (i.e., between non-English pairs)?

---

> ### Author Response · Authors · 2025-11-22
> **Response to Reviewer ErSk**
>
> Dear reviewer ErSk,
>
> Thank you for your review.
>
> # 1. The analysis of CoT remains superficial and the structural differences between MT and reasoning tasks
>
> Although we did not discuss why CoT is ineffective in a dedicated section, we do provide intuition throughout the paper. For example, in L462 we partially attribute the failure of “thinking” in MT to the scarcity of reasoning-like translation data on the internet. Such data is inherently limited: in translation, the primary artefact is the final output, whereas in mathematics the reasoning process itself is explicitly documented and valued. Our attempts of using a LLM (teacher) to build similar data (Section 5.2) and use them during CoTFT resulted in a failure to outperform IOFT probably because the teacher couldn't accurately build what it hasn’t seen.
>
> Moreover, in the context of RL for MT, the notion of a “verifiable reward” is not well-defined. The idea of a “correct translation” is far less absolute than a “correct answer” in mathematics (e.g. 1+1=2). As a result, we rely on proxy metrics such as BLEU or COMET (Section 6.2), but using these rewards does not reproduce the dramatic improvements observed in mathematics and code. We will clarify these points in the next version of the paper.
>
> Regarding model scale, we tested “thinking” vs. “not thinking” across a wide range of sizes (0.6B–32B, and 70B with DeepSeek-LLaMA-70B), to test the generality of our findings, and the conclusions held consistently at scale. Since fine-tuning (SFT) very Large LMs across our many scenarios is costly, we focused on medium-scale models (Gemma-3 4B and 1B for English→Xhosa/Lithuanian) and observed the same pattern. Additional SFT with Gemma-3-12B for English→Xhosa (via LoRA) confirms these findings. (Please refer to our answer to **Reviewer j3To** for the exact scores).
>
> We performed a focused comparison of “thinking” vs. “not thinking” for English→French in a reference-free setting using Qwen 3 32B. Annotators judged both outputs equally good or bad in 36% of cases, preferred the “thinking”  in 31%, and the “not thinking” model in 33%. We ran the same protocol for Turkish→English in a reference-based setting and found equal quality in 47% of cases; “thinking” was preferred in 30%, and “not thinking” in 23%. This validates the conclusion of our benchmarking at L.202.
>
> # 2. Grammatical Mistakes
>
> Thank you for pointing out these mistakes. We recognize that the sentence between L.350 and L.352 could be written in a clearer way. We meant “the best translation between those embedded in the thinking traces and the initial target translation”. In lines 416-417 there is a typo, **P** is meant for "Paraphrases" but **SP** is meant for "Syntactic Paraphrases" (not "Synthetic", as written in the paper, it is a typo as indicated by the titles on Figure 7). We’ll make sure to clarify this in the next version of the paper.
>
> # 3. Extreme scenarios where thinking might help
>
> Very small thinking models (<0.6B) are not widely available, and we already include models such as Qwen-0.6B in our evaluation. Moreover, Large LMs themselves struggle with low-resource languages (e.g., Wolof), so translating between 2 low-resource languages with an even smaller model would yield extremely low-quality outputs. In such conditions, the results would offer little meaningful insight, even by metric-based standards.
>
> However, we can resort to translating into “medium high-resource” or “mid-resource” languages such as Kazakh and Finnish or even high-resource languages like French. In Line 1812 (in Appendix B.1) we mention the fact that small LMs (Qwen-3-0.6B in particular) can struggle to consistently translate into some languages without thinking, because they fail to generate the correct target language from time to time. The absence of thinking makes them generate English, Chinese etc. but the issue quickly disappears as the models grow bigger. That is the reason why in Table 1, with Qwen-3-0.6B, you observe higher (better) BLEU scores w/ thinking than w/o thinking but bigger (worse) MetricX. This is because some translation attempts are in English and have better (smaller) metricX scores than real attempts in the target language, despite the metricX being more robust to this issue than alternative neural metrics. BLEU does not fall for this trap (this is also why we consider multiple metrics throughout the paper). To convince yourself, you can look at Table 16 (Appendix B.1) to see X to English directions and observe that Qwen-3-0.6B w/ thinking performs slightly better than w/o thinking with an agreement between both metrics.
>
> Regarding fine-tuning, we conducted our fine-tuning experiments on English to Xhosa and with `gemma-3-4b-pt` as the student. However, in Appendices B.3, B.4 and B.5 we find that fine-tuning `gemma-3-1b-pt` (much smaller) to build thinking MT models for English to Lithuanian helps us to reach the same conclusion (i.e. thinking is not very useful) despite it being very small.

---

### Official Review · Reviewer_j3To · 2025-10-31

**Soundness:** 2
**Presentation:** 3
**Contribution:** 3
**Rating:** 6
**Confidence:** 3

**Summary:**

This paper systematically investigates whether generating intermediate “thinking tokens” via LRMs/LLMs is beneficial for MT. The work benchmarks both “thinking” and “non-thinking” modes in SoTA LLMs across multiple languages and resource levels, performs distilled CoT fine-tuning with explanation templates inspired by human translation strategies, and examines a set of MT-specific prompting strategies for generating intermediate traces. Results reveal that neither thinking tokens nor CoT distillation improve translation quality compared to standard input-output fine-tuning, except when the intermediate traces involve actual translation attempts or parallel sentence pairs.

**Strengths:**

1.	The paper covers ten language directions across both high- and low-resource settings and evaluates on three MT benchmarks, providing strong empirical support for its main claims.
2.	The manuscript conducts a rigorous comparison between “thinking” and “non-thinking” modes using multiple LLMs under both zero-shot and few-shot scenarios.

**Weaknesses:**

1.	The paper clearly shows through extensive experiments that generic CoT reasoning does not improve MT, but provides little explanation as to why. It lacks discussion of possible factors such as model capacity, training signal limitations, or differences between reasoning and translation.
2.	The evaluation mainly relies on BLEU, MetricX, and other aggregate metrics, with limited analysis of error types, and qualitative comparisons between outputs with and without reasoning tokens or MT-specific traces.

**Questions:**

1.	The paper finds that MT-specific prompting strategies only help when they introduce actual drafts or partials into the intermediate trace. Can the authors clarify in quantitative terms what proportion of “traces” in each strategy (e.g., MAPS vs. CompTra) consist of direct translation attempts versus meta-reasoning? Are there cases where purely reasoning-based traces improved results without embedded translation data?
2.	How sensitive are the observed results to the choice of evaluation metric (e.g., BLEU vs. MetricX vs. COMET)? Did any metric reveal contrary trends with respect to reasoning token inclusion?

---

> ### Author Response · Authors · 2025-11-22
> **Response to Reviewer j3To**
>
> Dear reviewer j3To,
>
> Thank you for your review.
>
> # 1. Training signal limitations, differences between reasoning and MT,  model capacity
>
> Although we did not devote a full section to these factors, we provide intuition throughout the paper. For instance, in L.462 we partly attribute the failure of “thinking” in MT to the scarcity of reasoning-like translation data online. Unlike mathematics, where reasoning steps are explicitly documented, the translation process rarely includes intermediate rationale, making such data inherently limited. In RL for MT, the notion of a “verifiable reward” is also unclear: a “correct translation” is far less absolute than a numerical answer (1+1=2), so we rely on proxy metrics like BLEU or COMET (Section 6.2). These rewards, however, do not yield the large gains seen in math or code. We will clarify these points in the next revision.
>
> Regarding model capacity, we tested “thinking” vs “not thinking” across a wide range of sizes (0.6B–32B, and 70B with DeepSeek-LLaMA-70B), and the conclusions held consistently. Since fully fine-tuning very large models is costly, we focused on medium-scale models (Gemma-3 1B and 4B for English→Xhosa/Lithuanian) and observed the same pattern. Additional SFT with Gemma-3-12B for English→Xhosa (via LoRA) confirms these findings as shown below.
>
> ||IOFT|CoTFT-T3|CoTFT-CompTra|CoTFT-Refine|IOFT-Max(CompTra)|IOFT-Max(Refine)|IOFT-BoA
> -|-|-|-|-|-|-|-
> BLEU|15.4|13.0|16.6|16.0|15.1|18.8|**19.9**
> MetricX|7.0|8.9|7.1|7.2|7.5|5.6|**5.1**
>
> CoT distillation (Template T3) does not outperform IOFT, building intermediate information based on strategies such as CompTra, and Self-refine helps but ultimately, using them to refine the target side in the training dataset yields greater improvements. Unfortunately, scaling model size for RL does not fall within our capacity.
>
> # 2. Qualitative Comparison
>
> We agree that qualitative comparisons are valuable, but running them across all our setups was prohibitively expensive. Since the submission, however, we conducted a small human evaluation to complement our automatic results. With our available resources, we performed a focused comparison of “thinking” vs. “not thinking” for English→French using Qwen 3 32B. We sampled 100 FLORES-200 examples and ran reference-free pairwise ranking (better / worse / same) with native French speakers, excluding cases where the thinking model failed to produce a translation. Annotators judged both outputs equally good or bad in 36% of cases, preferred the “thinking”  in 31%, and the “not thinking” model in 33%. We ran the same protocol for Turkish→English in a reference-based setting and found equal quality in 47% of cases; “thinking” was preferred in 30%, and “not thinking” in 23%. Overall, these results support our statement in L.202: “MT capabilities of TM do not significantly benefit from thinking tokens.”
>
> # 3. Proportion of translation traces / positive results without them
>
> Here, meta-reasoning refers to parts of the trace that do not contain direct or partial translation attempts. The proportion of such content varies across strategies (Appendix A.4). Self-refine and CompTra contain only translation attempts (full or partial), whereas SBYS includes a research phase, TEaR has an MQM-style annotation step, and MAPS involves generating keywords, topics, and a demonstration. On average (measured in Gemma-3 tokens), non-translation content accounts for ~37% of SBYS traces, ~23% of MAPS, and ~16% of TEaR. Importantly, **a higher proportion of translation-related content does not guarantee better CoTFT performance** (Figure 4: CoTFT with MAPS > with CompTra or Self-refine).
>
> Stepwise strategies like MAPS and CompTra also make it straightforward to locate translation attempts, unlike free-form CoT (Section 5.1), where such attempts may appear implicitly (e.g., “his love of antiques”→“son amour …”, Figure 1). MAPS, for instance, has 7 steps applied to the source x, of which 3 (generation of keywords, topics, demonstration) are pure meta-reasoning and 4 are translation attempts (T0–T3). As shown in Figure 6, performing CoTFT with all steps (S0–S3, T0–T3) as intermediate information and **best-of(ground\_truth, T0-T3)** as target did not outperform IOFT using the same target, indicating that the meta-reasoning steps are not responsible for the gains. We performed the same analysis for all stepwise strategies and found no case where meta-reasoning without explicit translation attempts improved results.
>
> # 4. Sensitiveness to the evaluation metric
>
> In Table 3 (Appendix B.1), we also report chrF++ and XCOMET-XXL for our large-scale “thinking” versus “not thinking” comparison. Both metrics align with the BLEU and MetricX results, further reinforcing our conclusions about the limited usefulness of thinking in MT. We use MetricX24 as our primary metric because it correlates strongly with human judgments, while BLEU remains included for its intuitive, word-level interpretability for non-experts.

---

### Official Review · Reviewer_GfaJ · 2025-10-31

**Soundness:** 3
**Presentation:** 3
**Contribution:** 2
**Rating:** 6
**Confidence:** 4

**Summary:**

The paper studies whether thinking tokens (as intermediate natural language traces) actually help large reasoning models to translate better. There are two settings: comparing “thinking mode” at inference time versus forced “non-thinking” decoding, and comparing SFT that trains a student to “think, then translate” versus standard input-output fine-tuning. It also shows that using reasoning traces elicited through prompting methods that reflect the machine translation process, rather than just eliciting a CoT, boosts performance, but ultimately that the quality of the source-target pairs is most important, since IOFT using the best target continues to yield the highest results.

**Strengths:**

1. There is a clear (negative) result takeaway on the value (or lack thereof) of thinking tokens for MT, which is supported by a number of experiments and reasonable temperature sweeps
2. The method is easy to follow and reproduce, with the templates included in the appendix.
3. The analysis section (section 6) has some interesting insights, that decomposition is helpful (but ultimately, providing it as additional, augmented samples in IOFT is still most effective), and that a short RLVR run yields modest gains while preserving that the CoTFT warm-ups do not meaningfully help.

**Weaknesses:**

1. The evaluation scope is still dependent purely on automated metrics, and while the incorporation of a newer metric like MetricX-24 is promising, including some human evaluation to confirm quality of the targets, especially given usage of lower-resource languages, would be best.
2. The experiments performed are over single-language corpora. In addition to the current experiments, using a multi-task corpus (with multiple target languages) would be valuable to unveil whether there are any compositional effects present by performing CoT fine-tuning.
3. The duration of training, in both SFT and RL, is quite short. It is possible that certain models could require a longer training duration for the CoT format to be properly learned for this task, so ablations on training duration should justify whether this choice is optimal.

**Questions:**

1. For CompTra, any insights on whether phrase-level pairs help because they reduce the search space, or because they augment data?
2. It is reported in the paper that RL while rewarding just the target is insufficient, but could you generate synthetic CoTs to yield the (existing) target answer and reward that as well? 5k steps with 12 samples per prompt is still relatively small in terms of the number of prompts seen — do you have any intuitions of what would happen if you scaled this? How diverse are the generations, after each type of fine-tuning?

---

> ### Author Response · Authors · 2025-11-22
> **Response to Reviewer GfaJ (Part 1/2)**
>
> Dear reviewer GfaJ,
>
> Thank you for your review.
>
> # 1. Human Evaluation
> Given our available resources, we carried out a focused human comparison of “thinking” versus “not thinking” for English-to-French translation using Qwen 3 32B. We randomly sampled 100 FLORES-200 examples and carried out reference-free pairwise ranking (better, worse, same) with native French speakers. To avoid biasing the results against the thinking model, we excluded instances where it failed to produce a translation due to getting lost in its CoT. Annotators judged both translations as equally good or bad in 36% of cases; the “thinking” answers were preferred in 31% of cases, and the “not thinking” answers in 33%.  We conducted the same experiment in a reference-based fashion to evaluate Turkish-to-English and found that both translations are equally good or bad in 47% of the cases; “thinking” answers were preferred in 30% of cases, and the “not thinking” answers in 23%. Overall, these findings corroborate what we wrote in L.202: “ MT capabilities of TM do not significantly benefit from thinking tokens.”
>
> # 2. Multiple target languages at once
> If we understand correctly, the question concerns applying CoTFT to translate from English into multiple target languages simultaneously (e.g., English→French, German, and Italian). This is indeed an interesting direction, but it would require a dedicated, full-scale study beyond the scope of this paper. Our current work already covers 10 translation directions across 3 MT benchmarks using Qwen 3 models (which are multilingual models) that show no gains from thinking tokens in any of the settings we evaluated. As a result, we do not have the bandwidth here to further investigate cross-lingual transfer effects.
>
> # 3. Training duration
> We understand your concern about underfitting and argue that 5000 steps is enough for SFT given the size of our datasets. Two things matter here. First, the number of tokens seen per training step (it depends on the context length and the global batch size). Second, the number of tokens per training sample. Given these observations, we conducted preliminary runs and found that our hyperparameters (L.279) work well for IOFT. Here is the result of IOFT for 10K steps (we observe no significant gains beyond 5K steps and even before)
>
> steps (x1000)|1|2|3|4|5|6|7|8|9|10
> -|-|-|-|-|-|-|-|-|-|-
> BLEU|12.9|13.4|14.5|15.0|14.8|14.9|14.7|14.3|14.4|14.1
> MetricX|9.74|9.04|8.34|7.89|7.86|7.89|7.84|7.91|7.96|8.09
>
> For CoTFT, we use a 2048-token context length and 16 gradient accumulation steps to accommodate the larger training samples. As shown in Figure 4, there is no indication of underfitting: all strategies (especially self-refine and CompTra) plateau after roughly 15×200 steps. We studied the impact of gradient accumulation steps during CoT distillation with a fixed template and found little variation.
>
> Gradient Accumulation Steps|8|16|32
> -|-|-|-
> BLEU|14.1|**14.7**|14.5
> MetricX|8.65|8.33|**8.30**
>
> # 4. Scaling and Diversity
> We want to clarify that for RL, we use 4 GPUs (1 for vLLM), a "per device batch size" of 4 and 12 generations per prompt. We tried to scale this during IOFT+GRPO, doubling the value of both parameters. This resulted in small improvements.
>
> steps (x1000)|1|2|3|4|5|6|7|8|9|10
> -|-|-|-|-|-|-|-|-|-|-
> BS=4, 5K steps,12 generations per prompt
> BLEU|15.8|15.7|15.8|15.9|15.9
> MetricX|7.06|6.98|6.92|6.95|6.84
> BS=8, 10K steps, 24 generations per prompt
> BLEU|15.7|15.8|15.9|15.8|15.9|16.0|15.8|15.9|15.7|15.9
> MetricX|6.93|6.88|6.83|6.90|6.73|6.76|6.80|**6.66**|6.80|6.71
>
> As for diversity, we computed a pairwise BLEU similarity matrix across all fine-tuning settings to assess how similar their FLORES translations are at the word level. Across templates T1-T6, generations fall within 37-40 BLEU. The similarity between IOFT-Max(CompTra) and IOFT is 52.1 BLEU, which is expected given CompTra’s short phrases, as indicated in the paper (L.377–394). CoTFT outputs across the five prompting strategies show pairwise BLEU scores between 38 and 42, with MAPS and TEaR reaching up to 42, likely due to the embedded translation attempts that shape their outputs. IOFT-Max strategies follow the same trend. All these values remain moderate, consistent with our human evaluation, which showed substantial reformulation between “thinking” and “not thinking”.
>
> # 5. Insights on CompTra
> It is hard to be definitive, but we believe CompTra’s success likely stems from both factors you mentioned. The teacher’s short phrases effectively serve as additional supervision: although they are not used as training samples, they still help the model to learn the translation task by their formatting similar to few-shot examples. At the same time, they narrow the “answer space,” since their strong lexical overlap with the source encourages the model to rely heavily on their translations when producing the full sentence (similar to how similarity-based retrieval improves few-shot MT compared to random sampling).

---

> ### Author Response · Authors · 2025-11-22
> **Response to Reviewer GfaJ (Part 2/2)**
>
> # 6. Reinforcement Learning
>
> If we understand correctly, you suggest to use an LLM (e.g. DeepSeek-R1 etc. ) to generate intermediate tokens $IT_{target}(x, y)$ explaining how to obtain $y$ from $x$ for each $(x, y)$ of the parallel dataset used for RL and add an additional reward $Similarity(IT(x), IT_{target}(x, y))$? Such an experiment is too computationally expensive for us, as it would require generating data and running an additional RL fine-tuning stage. Nevertheless, we can share our intuition. Sections 3 and 5.2 already show that reasoning traces produced by LMs do not yield better MT performance. Fine-tuning a student to mimic such traces e.g. at the word level, is unlikely to help. In our CoTFT + RL experiments, we deliberately did not apply a reward to the CoT itself, aiming instead to see whether RL would naturally lead the model to adapt its CoT in useful ways. This did not happen. More broadly, determining what constitutes a good CoT for MT remains a research question in its own right. However, in the case of $IT(x)$ obtained with **CompTra**, we had already thought about evaluating the impact of adding an additional reward based on *BLASER-2.0 QE* scores evaluating the quality of the partial phrase-translation pairs within $IT(x)$.
>
> Steps (x1000)|1|2|3|4
> -|-|-|-|-
> *w/ internal reward*
> BLEU|18.2|17.9|18.2|18.2
> MetricX|5.93|5.89|5.90|**5.85**
> *w/o internal reward*
> BLEU|18.2|18.0|18.1|18.0
> MetricX|6.11|5.98|6.04|5.94
>
> Adding such a reward provides a small improvement (about -0.1 MetricX, probably because the new reward is neural-based), suggesting that rewarding intermediate information can be beneficial.

---

### Official Review · Reviewer_3r9b · 2025-10-31

**Soundness:** 3
**Presentation:** 2
**Contribution:** 3
**Rating:** 6
**Confidence:** 4

**Summary:**

The paper - LLM Reasoning for Machine Translation: Synthetic data generation over thinking tokens - presents some interesting results looking at the ability of cutting edge LLMs and common reasoning strategies in machine translation tasks. These results are interesting, especially in how they go against the prevailing trend. In general the authors find that reasoning, and CoT does not improve MT tasks. But that traditional translation prompting with examples, or the inclusion of additional translation examples in CoT does result in improved performance. This suggests to me that while translation tasks ought to be a strong use case for reasoning (applying a hierarchical set of rules) it remains data limited, and still best suited to a few-shot still approach. This is an important result in the use of LLMs for MT, but could also shine a light on the true reasoning abilities more generally.

**Strengths:**

- The paper is well written and really well structured, easy to read + navigate to sections to find clarifying details

- The breadth of evaluations and models used feels well principled, and by fine tuning goes beyond a simple evaluation of existing models

- The authors present three really clear experiments

    - The “thinking” tokens of the LRM does not improve the MT performance

    - Distilling the CoT outputs and using them as inputs for standard FT also doesn’t improve the MT performance

    - The evidence that real examples - and the number of them - remains the key differentiator in MT performance

- Comprehensive results presented (table 1) showing the widespread effect the authors describe - that thinking tokens do not meaningfully change the quality of results.

- The authors evaluate performance fine tuning the models - it is reasonable to think that the MT task may require a different training setup, and the authors explore this well with both traditional FT and using the CoT as part of the FT.

- The observation that synthetic data during inference time demonstrably improves results is a valuable insight, and I suspect this has utility beyond MT tasks.

**Weaknesses:**

- The RL section has I think one of the most interesting sentences of the paper - this feels like a really crucial observation - and I wonder if the authors have any thoughts on how to conduct an experiment to evaluate this? One could imagine a prompting strategy / finetuning data framed as iterative steps might map better onto the “reasoning” framework that works well in maths and programming cases?


“Notably, COTFT still does not outperform IOFT, even with RL. This is consistent with Zheng et al.’s (2025) findings, namely that CoT signals fail to induce meaningful reasoning when the reward is applied only to the final translation. Moreover, unlike mathematics where step-by-step explanations are widely present in pre-training corpora (proofs), it is not the case for translation data. This scarcity of reasoning-like data may explain CoT’s limited effectiveness in MT.” ([“LLM Reasoning for Machine Translation: Synthetic Data Generation over Thinking Tokens”, 2025, p. 9](zotero://select/library/items/2Y7C2KKT)) ([pdf](zotero://open-pdf/library/items/2URS85BT?page=9))

- There are additional metrics that evaluate translation - I’m thinking of evals like COMET - that give more of a sense of the impact on style and tone - I don’t think this is needs inclusion, but would offer insight into CoT / Reasoning on tone?

- There is a lack of qualative examples side by side? There are prompting strategies in the appendix, but it would be really useful to see some side by side examples to understand the way in which these translations work / do not work and the reasoning traces? I appreciate this is not always easy to present what can be quite long traces, but if possible I think it could merit inclusion.

- Personally I don’t particularly like the plots which show performance over the duration of training. I feel like they take up a lot of space, but don’t convey much useful information over alternative figures.

**Questions:**

- Would it be possible to conduct a quick experiment - just with a small subset of data - using prompts / fine tune data that show a step wise approach to translation as mentioned in Sec. 6.2?

    - I understand this would be a large study to do exhaustively - but could the authors comment on if they see value here?

- Could you consider a different presentation style for the results in Figs 2 - 7 ? As far as I can tell the argument isn’t about the training dynamics, so presumably the value of interest is the final score? Can you show this as a bar chart if the dynamics aren’t the most relevant? Or even as a table if that’s better?

    - Bar charts probably best ?

    - Or if the curves are important many one example of that and then the rest of the results summarized in a table?


Nitpicks

- Could the plotting of Figs 2- 7 be improved, the axes / labels are too small

- X axes on figures seem to be missing a label? 1 - 25 what? I know it’s described the first time, but this needs to be clear on each Fig.

---

> ### Author Response · Authors · 2025-11-22
> **Response to Reviewer 3r9b**
>
> Dear reviewer 3r9b,
>
> Thank you for your review.
>
> # 1.  Prompting strategy/fine-tuning data framed as iterative steps might map better onto the “reasoning” framework that works well in math and programming
>
> Most prompting strategies for MT are a succession of iterative steps, for example SBYS is a four-step process comprising pre-drafting research (identification of expressions that may pose a challenge for translation), drafting, refinement, and proofreading (for terminology, fluency, etc.). TEaR is a three-step process comprising translation (in a few-shot setting), annotation (of potential translation errors), and refinement (based on these annotations). For each strategy, we concatenate the teacher’s outputs for all steps (applied to the source) to form the intermediate information used for CoTFT with the student model. In an RL setting, however, it is unclear how to assign rewards to individual steps, for instance, how should we reward the identification of challenging expressions in SBYS, or the quality estimation stage in TEaR? Although translation attempts within the CoT can receive rewards, we ultimately reward only the final target translation, which is the key artefact in MT, hoping the model will implicitly learn to refine its reasoning to improve translation quality.
>
> # 2. COMET-based evaluation
>
> We agree that metrics that take into account tone/style are important. We chose to include MetricX for this reason, as it evaluates in a more semantic way than surface-based metrics such as BLEU in the same way as COMET, and it is also one of the main metrics used to produce the preliminary ranking of the most recent WMT25. It is also typically better correlated with human judgments than COMET. We nevertheless provided an evaluation of “thinking” vs “not thinking” using Qwen 3 models in Table 3 (Appendix B.1) with 2 additional metrics (XCOMET-XXL and chrF++) for transparency reasons, and they agree with BLEU and MetricX.
>
> # 3. Side-by-side qualitative comparison and inclusion in the paper
>
> We agree that providing side-by-side examples of translations for the prompting strategies help have a better sense of what they look like. We attempted to do so in Figure 1 but it is not as informative as fully-fledged examples. Given the additional page granted upon acceptance, we will examine the possibility of adding examples in the main part of the paper and if there is not enough space,  we will definitely at least add them to the appendix (just as we did with the prompts in Appendices A.3 and A.4). We have also performed human evaluation to confirm that thinking tokens do not help improve MT performance.
>
> # 4. Stepwise approach to MT
>
> By stepwise approach to MT, we meant reasoning-like data for MT, explaining step by step how a translator comes up with a translation for a sentence. Such data is inherently limited: in translation, the primary artefact is the final output, whereas in mathematics the reasoning process itself is explicitly documented and valued. In an attempt to generate such data, we prompted a teacher such as LLaMA-4-Scout (and DeepSeek-LLaMA in Appendix B.6) to produce what it “thinks” step-by-step MT reasoning looks like given a source-target pair. The prompt can be as simple as **T0**=*"Explain step by step how to translate the source sentence into the target sentence."* The aim of Section 5.2 was to see if data obtained in such a way can help as intermediate information (CoT distillation) when fine-tuning `gemma-4b-pt`. We considered 6 step-by-step templates (T1-T6; provided in Appendix A.3) and found that in all the cases the performances are on par with IOFT. We will clarify this in the paper.
>
> Here are the results of CoT fine-tuning (CoTFT) `gemma-4b-pt` with T0 (we can provide an example of a traces obtained with the teacher and T0 in a subsequent answer if you want to see what they usually look like, feel free to ask):
>
> Steps (x1000)|1|2|3|4|5
> -|-|-|-|-|-
> **IOFT**
> BLEU|12.92|13.18|14.43|**14.80**|14.64
> MetricX|9.63|9.20|8.34|**7.75**|7.97
> **CoTFT with T0**
> BLEU|12.05|13.62|13.75|14.31|14.70
> MetricX|10.44|9.23|8.86|8.33|8.33
>
> CoTFT with T0 fails to outperform IOFT.
>
> # 5. The layout of the plot, their labels and axes
> Thank you for your suggestion. We agree that these plots take a lot of space, however, we included them in this format because we believe that showing the training dynamic is important, in particular to make sure that there is no underfitting/overfitting. Typically, reviewer GfaJ’s concerns about us not training long enough can be addressed by using this type of plot. Moreover, in the case of RL, it can give a sense of when an “Aha moment” occurs.
>
> We’ll make sure to correctly label the x-axis across all figures and make the labels more visible for the readers. Across all figures, we plot the performance every 200 steps, 1-25 corresponds to 1x200 steps to 25x200=5000 steps (2-50 is better if we specify x100 as label). We’ll make this clearer in the next version of the paper.

---

### Author Response · Authors · 2025-11-28
**Comment to all the reviewers (summary of the revisions)**

We thank the reviewers for their thoughtful feedback and for the opportunity to improve our work. In response, we have uploaded a revised version of the paper (with new content highlighted in blue) with the following updates:

### **1. Writing Improvements** (**Reviewer 3r9b** and **Reviewer ErSk**)

- We corrected typos and added clarifications throughout the paper.
- In Sections 2 and 4, we now clearly describe each prompting strategy used in this work.
- Appendix C.2 includes example outputs for all CoT distillation templates and MT prompting strategies.
- All figures now include step counts on the x-axis, and legends have been expanded for readability.
- We incorporated additional insights on the marginal utility of thinking during GRPO in Section 6.2 and expanded the discussion in the conclusion.

### **2. Human Evaluation (Appendix B.2)** (**Reviewer GfaJ** and  **Reviewer j3To**)

- We added human evaluation results obtained with `Qwen3-32B`.

### **3. Impact of Model Size: Larger Student Model (Appendix B.10)** (**Reviewer GfaJ**)

- We added SFT experiments with a larger student model (`gemma-3-12b-pt`) to verify that our findings for **SFT** hold at a larger scale. Experiments include **CoTFT** with all six CoT distillation templates, **CoTFT** with traces from all five prompting strategies, and **IOFT-Max** for each strategy.

### **4. Scaling Training (Appendix C.1)** (**Reviewer GfaJ** and **Reviewer j3To**)

- We studied the effect of longer training for both SFT (**IOFT** and **CoTFT**) and RL (**IOFT**). We also investigated an additional reward term for GRPO when starting from a **CoTFT with CompTra** model.

### **5. Generation Diversity (Appendix C.1)** (**Reviewer GfaJ**)

- We added a similarity matrix reporting BLEU scores between FLORES-200 outputs for every pair of strategies evaluated in the paper.

We believe these changes address the reviewers’ main concerns and strengthen the overall contribution. We welcome further discussion and feedback.

**Note**: Some concerns had already been addressed by results in the appendix, particularly those regarding metric choice and evaluation with COMET-based metrics.

---

### Meta-Review · Area_Chair_rM46 · 2026-01-07

**Summary:**

Across reviewers, the central concern was whether explicit reasoning or chain-of-thought (CoT) signals meaningfully improve machine translation (MT) beyond standard input–output fine-tuning. Reviewers generally agreed that the paper provides a well-supported negative result: generic “thinking tokens” and CoT distillation do not improve MT, while gains arise only when intermediate traces contain actual translation attempts or higher-quality parallel data. Remaining doubts focused on (i) the depth of explanation for why CoT fails in MT compared to math/code, (ii) reliance on automated metrics and limited qualitative analysis, and (iii) whether alternative training regimes (different RL rewards, multitask or extreme low-resource settings) might change the conclusion.

**Reviewer Concerns:**

The rebuttal provides targeted human evaluations and additional metrics and ablations, and clarifies several experimental details.

The remaining concerns or partially addressed concerns include:
- Depth of conceptual explanation (j3To, ErSk): While intuition is now clearer, some reviewers may still find the analysis of structural differences between MT and reasoning tasks less developed than desired.
- Qualitative examples and presentation (3r9b): Promised additions (side-by-side examples, improved figures) are deferred to the camera-ready version.
- Broader generalization (GfaJ): Multitask or more radical RL setups remain unexplored due to cost, leaving open (but speculative) avenues.

**Reviewer Scores:**

While the rebuttal may have addressed some concerns regarding evaluation validity (human evaluation), training duration, and stepwise prompting, the remaining concerns seem critical to the evaluation of this work. In particular, a more in-depth analysis of the structural differences between MT and reasoning tasks is required. Also, exploring whether alternative training regimes, e.g., different RL rewards, multitask or extreme low-resource settings, might change the conclusion is also critical. Qualitative analysis and comparison should also be presented.

---

### Decision · Program_Chairs · 2026-01-26

Reject